

**Exploring the potential relationship between the occurrence**
**of debris flow and landslide**
**Zhu Liang[1], Changming Wang[1], Donghe Ma[2] and Kaleem Ullah Jan Khan[1]**
[1]College of Construction Engineering, Jilin University, 130000 Changchun, People's Republic of
China;
[2]China Water Northeastern Investigation, Design and Research Co.Ltd.
E-mail:wangcm@jlu.edu.cn
**Abstract:** The aim of the present study is to explore the potential relationship between debris flow
and soil slide by establishing susceptibility zoning maps (SZM) separately with the use of random
forest. Longzi County, located in Southeastern Tibet, where historical landslides occurred
commonly, was selected as the study area. The work has been carried out with the following steps:
**(1)** An inventory map consisting of 448 landslides (399 soil slides and 49 debris flows) was
determined; **(2)** Slope units and 11 conditioning factors were prepared for the susceptibility
modelling of landslide while watershed units and 12 factors for debris flow; **(3)** SZM were
constructed for landslide and debris flow, respectively, with the use of random forest; **(4)** The
performance of two models were evaluated by 5-fold cross-validation using relative operating
characteristic curve (ROC), area under the curve (AUC) and statistical measures; **(5)** The potential
relationship between soil slide and debris flow was explored by the superimposition of two zoning
maps; **(6)** Gini index was applied to determined the major factors and analyze the difference
between debris flow and soil slide; **(7)** A combined susceptibility map with two kinds of disaster



was obtained. Two models had demonstrated great predictive capabilities, of which accuracy and
AUC was 87.33%, 0.902 and 85.17%, 0.892, respectively. The loose sources need by the debris
flow were not necessarily brought by the landslides although most landslides can be converted
into debris flow. The area prone to debris flow did not promote the occurrence of landslide. A
susceptibility zoning map composed of two or more natural disasters is comprehensive and
significant in this regard, which provides valuable reference for researches of disaster-chain and
engineering applications.
**Key words:** Landslide; Debris flow; Susceptibility; Random forest; Potential relationship

# 31    1. Introduction

Soil slide and debris flow are two kinds of natural phenomenon mainly occurring in mountainous
areas, which pose considerable threats to people, industries, and the environment directly or
indirectly. Generally, damages can be decreased to a certain extent by predicting the likely
location of future disasters (Pradhan, 2010). Thus, extensive research has been conducted for the
prediction and susceptibility assessment of soil slide and debris flow.

In geomorphology, a "landslide" is the movement of a mass of rock, debris or earth down a

slope, under the influence of gravity (Cruden and Varnes, 1996). According to different variables,
landslides can be divided into different types (Varnes, 1978). Debris flow is a specific type of
landslide, which can be defined as (Hungr et al. 2013): ''Very rapid to extremely rapid surging
flow of saturated debris in a steep channel''. Generally, slides that occur on a steep slope and
become disaggregated as they tumble down can transform into debris flows if they contain



sufficient water for saturation (Huang et al., 2020). Therefore, slides may provide sufficient
material source for the occurrence of debris flow and most of the slides are accompanied by debris
flow. In the past, few scholars have specifically distinguished the slides and debris flow in terms
of susceptibility assessment (Alessandro et al., 2015; Guzzetti et al., 2005). In addition, some
scholars made separate evaluations of slides and debris flow (Park et al., 2011; Haydar et al.,
2016). Some scholars have proposed a coupled model of landslide-debris flow (Chiang et al., 2012;
Gomes et al., 2013). However, not every slide has evolved into a debris flow and the material
source of the debris flow is not necessary coming from slides. The formation and manifestations
of different types of landslides are different, especially debris flow, which is a kind of "wet
flow"(Varnes, 1978). In other words, there is no determined connection between debris flow and
other types of landslide. Therefore, the potential relationship between debris flow and other types
of landslide need further exploration.

Besides, the conditioning factors and mapping units involved in the susceptibility assessment

different kinds of landslides are not identical. Especially slope and water content are the most
critical factors controlling movements of debris flow (Takahashi 2007). Therefore, it is more
reasonable to evaluate the susceptibility of different kinds of landslides separately. As an example,
one landslide inventory map includes only one type of landslide, as does debris flow.

The methods of susceptibility assessment can be broadly classified as qualitative or

quantitative (Aleotti et al., 1999). Several methods and approaches have been proposed and tested
to ascertain susceptibility, such as physical-based approaches (Carrara et al., 2008), heuristic
methods (Blais et al., 2016) and statistically-based approaches (Reichenbach et al., 2018). In
addition, new machine learning models, such as neural networks (Park et al.,2013), support vector




machines (Colkesen et al.,2016) and random forest (RF) (Zhu et al., 2020a), have also been
applied.

The Longzi County in Southeastern Tibet is always exposed to slides and debris flow hazard

because of climatic and topographic conditions, which is chosen as the study area. The purpose of
the present study is to explore the potential relationship between the occurrence of debris flow and
soil slide by establishing susceptibility zoning maps separately with the use of random forest. It
also provides a reference for the study of landslide-debris flow, a common disaster chain.

## 72    2. Materials

## 73    2.1  Study area

The study area located in Longzi Township, Longzi County, Southeastern Tibet is bounded by
longitudes of 92°15'E and 92°45'E, latitudes of 28°10'N and 28°30'N (Fig.1). It covers an area of
about 535 km$^2$ with a population of more than 6000. The study area belongs to a semi-arid
temperate monsoon climate with the annual rainfall of 279 mm, mainly concentrated in May to
September. The seismic intensity within the area has a degree of VIII on the modified Mercalli
index.

The study area belongs to the zone of stratigraphic division of the Northern Himalayan block.

The strata is mainly composed of Mesozoic Cretaceous, Jurassic, Triassic, and Cenozoic units.
There were three common lithology observed during our field investigation: Siltstone from the
Laka Formation ($K_1l$); Conglomerates from the Weimei Formation ($J_3w$) and Quaternary slope
wash ($Q_4^{el+dl}$) from the Cenozoic strata.

The disasters in the study area mainly consist of rain-fed high frequency debris flows and


landslides, which destroyed and flooded roads, bridges, farmlands, villages, etc., causing great
economic losses.

## 2.2  Landslide and debris flow inventory

The statistically-based susceptibility models are based on an important assumption: future
landslides will be more likely to occur under the conditions which led to the landslides past and
present (Varnes, 1984; Furlani and Ninfo, 2015). Therefore, a complete and accurate inventory
map is the key for model training and validation. In this study, data comes from historical records,
field surveys (**Fig.2 and Fig.3**) and interpretation of Google Earth images carried out in Google
Earth pro 7.1(**Fig.4**). Finally, a total of 399 soil slides and 49 debris flow locations were recorded
and mapped (**Fig.1**).

## 2.3  Mapping units

The selection of the mapping unit is an important pre-requisite for susceptibility modelling
(Guzzetti, 2006). The main mapping units commonly used for landslide and debris flow
susceptibility assessment are grid cells (Reichenbach et al., 2018). Despite its popularity and
operational advantages, grid-cells have clear drawbacks for susceptibility modelling (Guzzetti et
al., 1999). There is no physical relationship between a grid-cell and slope, while slope units can
make up for this deficiency. Depending on the landslide type, a slope unit may correspond to an
individual slope, an ensemble of adjacent slopes or a small catchment (Reichenbach et al., 2018).
The geometry of debris flow is better represented by a polygon or a set of polygons in vector
format. In the present study, adjacent slope units were applied to the susceptibility assessment of
soil slide. First-order sub-catchments, which is also called watershed unit, was applied to the



susceptibility of debris flow (Francesco et al., 2015; Zhu et al., 2020b). Accordingly, the study
area was divided into 1003 slope units for the modeling of soil slide or 174 watershed units for
debris flow.

## 2.4  Controlling factors and mapping

The selection of evaluation parameters is another key prerequisite to ensure that the model is
accurate and reasonable. With reference to previous studies (Ahmed et al., 2016; Xu et al., 2013;
Braun et al., 2018), there are differences in the controlling parameters used in soil slide and debris
flow susceptibility assessment. The occurrence of debris flow emphasizes the indispensability of
provenance, topography and triggering factors. Availability, reliability, and practicality of the
factor data were also considered (van Westen et al., 2008). In this paper, 11 controlling factors are
selected for the susceptibility assessment of landslide, including distance to fault, distance to road,
distance to river, annual rainfall, slope angle, aspect, plan curvature, profile curvature, topographic
wetness index, elevation and maximum elevation difference. Besides, a total of 12 controlling
factors, including basin area, main channel length, normalized difference vegetation index (NDVI),
drainage density, roundness, melton, average gradient of main channel, slope angle, maximum
elevation difference, annual rainfall, distance to fault and elevation were selected to fully reflect
the characteristics of the watershed for the susceptibility assessment of debris flow. Detailed
information on conditioning factors is shown in **Fig.5a~5m**. A brief description of each controlling
factor is given below.
Aspect, which is frequently used as landslide controlling factor (Dai and Lee, 2002), was
reclassified into 8 classes (**Fig. 5g**). Plan curvature and profile curvature were both considered and





reclassified into six classes (**Fig. 5b and 5e**). Generally, faults, rivers and roads play a key role in
the occurrence of landslides and were reclassified into seven classes using an interval of 1500m
(**Fig. 5i~k**). Topographic wetness index was reclassified into five classes (**Fig. 5h**).

NDVI reflects the vegetation conditions in the area and was reclassified into 5 classes(**Fig.

6b**). Drainage density is the ratio of the total drainage length to the watershed area and was
reclassified into six classes (**Fig.6 g**). Roundness refers to the ratio of the area of a basin to the
area of a circle with the same circumference and was reclassified into six classes (**Fig.6 d**) .
Melton ratio refers to the ratio of the degree of undulation in the watershed to the square root of
the arithmetic area of the watershed (Melton, 1965), which is reclassified into seven classes (**Fig.
6a**). Considering the correlation between the two controlling factors, basin area and main channel
length are represented by the same graph, which was reclassified into four classes (**Fig.6h**).
Average gradient of main channel, which is the ratio of the maximum elevation difference of main
channel to its linear length, was reclassified into six classes (**Fig. 6j**).

Rainfall is the only triggering factor to be considered for both landslide and debris flow in this

paper, which was reclassified into six classes (**Fig. 5a and Fig. 6c**). Slope angle is frequently employed
in both landslide and debris flow susceptibility mapping and was reclassified into six classes (**Fig. 5f
and Fig. 6i**). Maximum elevation difference reflects the kinetic energy condition and is reclassified
into 6 classes using an interval of 200m (**Fig. 5c and Fig. 6e**). Elevation was reclassified into five
classes (**Fig. 5d and Fig. 6f**), which has also been used by many authors (Ayalew and Yamagishi, 2005;
Pourghasemi et al. 2013a, b) .

Totally 18 factors are obtained by processing the row data in the ArcGIS 10.2 platform.

Morpholigical and topographic related factors were derived from the DEM with a resolution of 30

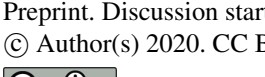



× 30 m. Geological related factors were extracted from 1:50000 geological maps. Rainfall is one
of the most important external factors inducing landslides and debris flow, which was determined
by ordinary kriging interpolation in ArcGIS by collecting data of 11 precipitation stations near the
area under study as a reference.

## 3. Methods


### 3.1 Sampling strategy and performance assessment


Statistical models for landslide susceptibility zoning reconstruct the relationships between
dependent and independent variables using training sets, and verify these relationships using
validation sets (Guzzetti et al., 2006a,b), which usually implies the partitioning of the inventory in
subsets. The sampling strategy affects the results of the susceptibility map (Yilmaz, 2010). Based
on temporal, spatial or random criteria, the partition of landslide inventories can be made (Chung
and Fabbri, 2003) and the most applied one is a one-time random selection (Reichenbach et al., 2018).
However, there is a need for a more reliable estimation of the model performance. The ability of
the models to classify independent test data was elaborated using a k-fold cross validation
procedure (k=5 in this paper) (James et al., 2013).

The computation of the area under the curve (AUC) is the most popular metrics to estimate

the quality of model , which has been applied for ROC curves( Green and Swets, 1966). It is one
of the most commonly used indicators. Three statistical metrics as accuracy, sensitivity, and
specificity are generally applied to assess the performance of the landslide susceptibility models
(Tien Bui et al. 2016).
$$Accuracy = \frac{TP+TN}{TP+TN+FP+FN}$$






$$Sensitivity = \frac{TP}{TP+FN}$$

$$Specificity = \frac{TN}{FP+TN}$$

(1)

where True Positives (TP), i.e., cells predicted unstable and observed unstable, True Negatives
(TN), i.e., cells predicted stable and observed stable, False Positives (FP), i.e., cells predicted
unstable but observed stable and False Negatives (FN), i.e., cells predicted stable but observed
unstable.

## 3.2 Random Forests

Random forest (RF) is a powerful ensemble-learning method and was first introduced by Breiman
(2001). RF uses the bagging technique (bootstrap aggregation) to select, at each node of the tree,
random samples of variables and observations as the training data set for model calibration.
Unselected cases (out of bag) are used to calculate the error of the model (OOB Error). The
increase in OOB error is proportional to the importance of the predictive variable (Breiman and
Cutler 2004). There are no restrictions on the types of variables, either numerical or categorical.
RF has the ability to reduce errors caused by unbalanced data, which is suitable for susceptibility
assessment.
In order to obtain reliable results of non-parametric models, their respective
hyperparameters must be optimized before application (Schratz et al., 2019). Scikit-learn package
(Pedregosa et al.,2011) in the programming software python version 3.7 was used for the
modeling. The number of trees    and the number of predictive variables used to split the nodes are
two user-defined parameters required to grow a random forest (Ahmed et al.,2016). The involved



parameters for modeling utilized in this study were shown in **Table 1.** Gini index (the larger the
value of the obtained result, the greater the contribution to the occurrence of landslide)
(Breiman,2001) was applied to analyze the major conditioning factors for both soil slide and
debris flow.

# 4. Results and verification

## 4.1  Landslide susceptibility mapping results

The predictive accuracy, ROC curves and AUC values of the RF model using training data were
showed in **Table 2** and **Fig. 7**. The RF model ensured a satisfactory performance of for classifying
landslides with sensitivity value of 91.62%. In terms of the classification of non-landslides zones,
specificity value also reached 89.06. An AUC equals to 1 indicates perfect prediction accuracy
(Vorpahl et al., 2012). The RF model had great performance in terms of AUC, with value of 0.976.
Standard error (St.), confidence interval (CI) at 95% and significance (Sig.) were applied as three
evaluation statistics. All these results indicated a reasonable goodness-of-fit for models with the
training dataset, for which the values were reasonably small.

Verifying the generalization ability of the model is a key step in prediction models as shown

in **Table 3** and **Fig. 7**. Accordingly, the values of sensitivity and specificity were 88.69% and
86.05%, respectively. The model also achieved a great performance in terms of AUC with value of
0.902. In comparison with the training model, the accuracy and AUC values have slightly
decreased, but still perform well.

The landslide susceptibility map was reclassified into five classes: very low (0~0.2), low

(0.2~0.4), moderate (0.4~0.6), high (0.6~0.8), very high (0.8~1) by using the equal spacing


method (**Fig.8** ). The map should satisfy two spatial effective rules: (1) The existing disaster points
should belong to the high-susceptibility class and **(2)** The high-susceptibility class should cover
only small areas (Bui et al. 2012). The number of units belonging to very high class reached 179,
accounting for 17% (**Fig.9**). Disaster points were mostly in the dark (red or orange) areas. The
units belonging to moderate class accounted for the smallest proportion, at 13% (**Fig.9**).

The controlling factors with significant effects were selected and normalized as shown in

**Table 2**. The weight values of slope angle, distance to fault, plan curvature and topographic wetness
index was 0.21, 0.19, 0.17, 0.13 respectively, which was closely related to the occurrence of
landslide. The weight values of distance to road, maximum elevation difference, profile curvature
and elevation are less than 0.1 as 0.08, 0.08, 0.06, and 0.05, respectively (**Fig.10**).

## 222   4.2 Debris flow susceptibility mapping result

The debris flow susceptibility model perform well with a very high sensitivity and specificity
values as 87.80% and 88.89%, respectively. In terms of accuracy and AUC, the model had also a
great prediction performance with the value of 88.57% and 0.967 (**Fig.7**). Three evaluation
statistics also indicate a reasonable goodness-of-fit for the model.

**Table 3** shows that the values of sensitivity and specificity were 85.71% and 84.62%, which

were slightly decreased compared to the training model. However, the model had achieved a great
performance in terms of AUC, with value of 0.892.

The number of units belonging to very high-class reached to 26, which was accounting for

15% while the units belonging to high-class accounted for the smallest proportion at 13%. More
than half of the units (58%) belong to on a low or very low-class (**Fig.9**). Disaster points were





mostly in the dark (Bright or deep red) areas (**Fig.8**).

The weight values of main channel length, roundness and slope angle were 0.25, 0.16, 0.14

respectively, which has significant influence on the occurrence of debris flow. The weight values
of elevation, maximum elevation difference, melton and basin area are close to 0.1, which are 0.13,
0.12, 0.1, and 0.1 respectively(**Fig.10**).

## 238 **4.3 Analysis and comparison of landslide and debris flow**
## 239 **susceptibility**

It is worth comparing the two susceptibility zoning maps. In terms of prediction accuracy, the
values of sensitivity, specificity and AUC of landslide model were slightly higher than that of
debris flow. However, both models achieved high predictive performance. Therefore, the landslide
and debris flow susceptibility assessment models based on RF are reliable. The purpose of the
present study is to explore the potential relationship between landslides and debris flows by
establishing the respective susceptibility zoning maps. Figure 11 shows the overlapping areas
between debris flow and landslide in high or very high-class of susceptibility zoning map. It can
be seen that most of the areas with high or very high-class in the map of debris flow are covered
with landslides. However, there are also non-overlapping areas between the two zoning maps.
There are 23 watershed units belonging to high-class in the debris flow susceptibility zoning map
(**Fig.8**), of which 17 units are covered with high or very high-class slope units in the landslide
zoning map (**Table 5**). In addition, there are 4 watershed units covered with low or very low class
slope units. In the same way, 19 watershed units belonging to very high-class are covered with
high or very high-class slop units and 4 watershed units with low or very low-class slop units. In



other words, more than 70% of the high or very high-class watershed units are covered with high
or very high-class slope units. However, there are still 30% of watershed units with high or very
high-class without the distribution of slope units in corresponding grades. It validated the previous
view that most of landslides can be transformed into debris flows. Factor analysis was applied to
further analyze the reasons for the difference. 36 watershed units with distribution of high or very
high-grade slope units were taken as model 1 and the left 8 watershed units as model 2 (**Table 5**).
The KMO (Kaiser-Meyer-Olkin) and significance (Sig.) testing are two statistical parameters
which ensured the feasibility before application. The KMO values were 0.766 and 0.643
respectively, which indicated that the correlation between variables was obvious and suitable for
factor analysis (**Table 6**). In model 1, the cumulative contribution rate of the first three factors (C1,
C2 ,C3 ) reached to 83.6%, while the cumulative contribution rate of the first four factors (F1,
F2 ,F3 and F4 ) reached to 80.5% for model 2 (**Table 7**). According to the correlation coefficient
of each common factor (**Table 7**), the first common factor mainly highlighted the information of
basin area, main channel length and maximum elevation difference. Similarly, the second and the
third common factor highlighted the information of slope angle and elevation and roundness,
respectively. The difference between the two models is that the second model has the fourth
common factor (**Table 8**), which emphasized the effects of rainfall and distance to the fault. The
transformation from a landslide to a debris flow often occurs during heavy rainfall (Takahashi,
1978), and the landslides are the source area. But landslides are not the only source of debris flows.
The loose material distributed in the basin is not necessarily caused by landslide.
In turn, we analyze the distribution of high or very high-class slope units in watershed units.
The landslide zoning map was put at the bottom floor and the debris flow zoning map on the top



floor (**Fig. 11**). There are 167 slope units belonging to high-class, of which 68 units (accounting
for about 40%) are distributed in the area of high or very high-class watershed units in the debris
flow zoning map (**Table 9**). Besides, 69 slope units (accounting for about 41%) are distributed in
the area of low or very low-class watershed units. Similarly, 53 slope units (accounting for about
30%) belonging to very high-class are distributed in the area of high or very high-class watershed
units and 88 slope units (accounting for about 50%) in low or very low-class slop units (**Table 9**).
Comparing with the extent of the landslide affecting the debris flow, the impact of the debris flow
on the landslide is not obvious. It indicated that the area prone to debris flow does not promote the
occurrence of landslides.
Finally, we took the center of gravity of 1,003 slope units as the potential hazard points and
spread them over 174 watershed units. Thus, a combining susceptibility zoination map for
landslide and debris flow was obtained (**Fig.11**). The darker the color, the higher the class of
susceptibility will be. It can be seen that the susceptibility in the south is generally higher than that
in the north, and the area in the southwest is disaster-prone. The northeast and central locations in
the area are less likely to be affected by landslides and belong to low-susceptibility areas. Green or
yellow dots, which refer to slope units with very low or low- class in the landslide zoning map,
mainly distributed in light-colored areas but there are also quite a few green or yellow dots
distributed in dark areas, which means that the occurrence of debris flow not necessarily depend
on landslides. Blue or black spots are mainly distributed in dark areas but there are also quite a
few blue or black spots distributed in dark light areas, which means that landslide is not the only
condition for debris flow to occur. Most of the watershed units are distributed with two or more
colored dots, which means that there would be multiple slope units with different susceptibility


class in the same watershed. According to the combining susceptibility zoning map of landslide
and debris flow, the study area can be divided into 4 categories: **(1)** Low or very low-class
watershed units coupled with low or very low-class slope units; **(2)** Low or very low-class
watershed units coupled with high or very high-class slope units; **(3)** High or very high-class
watershed units coupled with low or very low-class slope units; **(4)** High or very high-class
watershed units coupled with high or very high-class slope units. We assume that the occurrence
of landslides can bring rich sources of debris flow, thereby promoting or aggravating the outbreak
of debris flow, that is, forming a landslide-debris flow disaster chain. Therefore, the susceptibility
assessment of the landslide-debris flow chain in the study area can be roughly divided into three
classes, which are low, moderate and high (**Table 10**).



## 5. Discussion

### 5.1 Method used for modeling

Many researchers have used different statistically-based methods to evaluate the susceptibility of landslides or debris flows. Logistic regression and discriminant analysis are the most popular methods to use in traditional multivariate statistical analysis. The performance of new learning machines, such as support vector machines and neural networks, has also been verified. RF, as a newly integrated learning machine, has less application in landslide and debris flow analysis. Actually, RF have powerful data processing capabilities and can simultaneously solve problems such as high-dimensional, unbalanced and data loss, which are common in geological disaster assessment. Most importantly, RF can compare the important differences between features and have ability to reduce errors caused by unbalanced data and, which achieved strong generalization properties (Zhu et al., 2020a).

### 5.2 Potential relationship between landslide and debris flow

There is a certain similarity in the evaluation of the susceptibility of landslide and debris flow from the concept, the selection of controlling factors and the application of modeling strategies. Therefore, some researchers have neglected the difference between landslide and debris flow i.e to express two different disasters with the same susceptibility zoning map (Ciurleo et al., 2016; Ciurleo et al., 2017; Persichillo et al., 2017;). However, similarity does not always mean consistency. Many researchers have previously conducted studies into the debris flow mobilization from shallow landslide using a coupled methodology. They are interested in the dynamic simulation of debris flow based on the prediction of landslide susceptibility (Wang et al., 2013;





Fan et al., 2017). However, not every landslide evolves into a debris flow, which means that the
analysis process is highly selective or uncertain. In the same way, the source of the debris flow is
not limited to landslide. There, the potential relationship between landslide and debris flow needs
to be discussed more reasonably and effectively. In this study, the corresponding influencing
factors and mapping units are selected to establish landslide and debris flow susceptibility zoning
maps, respectively. The potential relationship between landslide and debris flow is explored in two
ways: **1)** Superimposing the high or very high-class susceptibility areas in the two maps; **2)**
Transforming the slope units into points and distributed them on the watershed units. The
relationship between landslide and debris flow is illustrated by the distribution of slope units of
different grades on the watershed units with different prone grades.

## 5.3 Necessity and feasibility of combining multiple natural disaster susceptibility zoning maps

Previous studies on susceptibility zoning mapping of disaster have agreed that one disaster
corresponds to one map. Multiple disasters may be bred simultaneously in a watershed unit and it
will cause some confusion in practical. For example, the probability of a disaster occurring in a
watershed is negligible, while it is high of another disaster. Therefore, we need to combine
multiple zoning maps at the same time to give a comprehensive evaluation, which is arduous to
achieve. On the one hand, the prediction accuracy and error of different zoning maps should be
similar or even consistent. On the other hand, the dimensions of the mapping unit should be
consistent or complementary. The fact that the appropriate prediction method and mapping units
applied to the two disasters makes it possible to merge the two zoning maps. Disaster risk is



higher in landslide-debris flow chain, causing significant loss of life and property. Therefore, two
natural disasters with potential relationship are simultaneously reflected in the same susceptibility
zoning map, which can better guide the implementation of engineering, such as landslide-debris
flow disaster chain.

## 6. Conclusion

In this study, susceptibility assessment models for landslide and debris flow are established
through RF, respectively and the performance of the models are excellent in terms of accuracy and
goodness of fit. The potential relationship between landslide and debris flow is discussed by the
superimposition of two zoning maps and the following conclusions can be drawn:
(1) The landslide and debris flow susceptibility assessment models based on random forest have
great performance of accuracy and goodness-of-fit and have the ability to analyze the relative
importance of different impact factors, which is suitable for the evaluation of natural disasters;
(2) Although most landslides will be converted into debris flow, the landslides are not necessarily
the source of debris flow, and the loose sources carried by the debris flow are not necessarily
brought by the landslides;
(3) By comparing the extent of the landslide affecting the debris flow, the impact of the debris
flow on the landslide is not obvious, which indicates that the area prone to debris flow does not
promote the occurrence of landslides;
(4) A susceptibility zoning map composed of two or more natural disasters is more
comprehensive and significant, which provides valuable reference for researchers and engineering
applications.




**Data availability.** The data used to support the findings of this study are included within the

article.

**Author contributions.** ZL was responsible for the writing and graphic production of the paper.

CMW was responsible for the revision of the paper. DHM was responsible for calculation. KUJK

was responsible for the translation.

**Competing interests.** The authors declare that they have no conflict of interest.

**Special issue statement.** This article is part of the special issue"Resilience to risks in built

environments". It is not associated with a conference.

# Acknowledgements

This work was supported by the National Natural Science Foundation of China (Grant No.

41972267 and 41572257 ).

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

**Table 1** The optimized parameters of RF

| Methods | Parameters |
| --- | --- |
| RF | Number of iterations, 100; number of execution slots, 10; 1oob_score = true; percentage of bag size, 0.382; max_features, sqrt; n_estimators, 500 |

**Table 2** Models' performance using training dataset

| Metrics | Landslide | Debris flow |
| --- | --- | --- |
| TP(%) | 88.71 | 87.80 |
| TN(%) | 91.89 | 88.89 |
| FP(%) | 11.29 | 12.20 |
| FN(%) | 8.11 | 11.11 |
| Sensitivity(%) | 91.62 | 88.77 |
| Specificity(%) | 89.06 | 87.93 |
| Accuracy(%) | 90.65 | 88.57 |





| | | |
|---|---|---|
| AUC | 0.976 | 0.967 |

**Table 3** Models' performance using verification dataset

| Metrics | Landslide | Debris flow |
|---|---|---|
| TP(%) | 85.56 | 85.71 |
| TN(%) | 89.09 | 84.62 |
| FP(%) | 14.44 | 14.29 |
| FN(%) | 10.91 | 15.38 |
| Sensitivity(%) | 88.69 | 84.79 |
| Specificity(%) | 86.05 | 85.55 |
| Accuracy(%) | 87.33 | 85.17 |
| AUC | 0.902 | 0.892 |

**Table 3** Variables importance assigned for landslide

| Test group | Slope angle | Distance to fault | Plan curvature | Topographic wetness index | Distance to road | Maximum elevation difference | Profile curvature | Elevation |
|---|---|---|---|---|---|---|---|---|
| Landslide | 0.21 | 0.19 | 0.17 | 0.13 | 0.08 | 0.07 | 0.06 | 0.05 |

**Table 4** Variables importance assigned for debris flow

| Test group | Main channel length | Roundness | Slope angle | Elevation | Maximum elevation difference | Melton | Basin area |
|---|---|---|---|---|---|---|---|
| Debris flow | 0.25 | 0.16 | 0.14 | 0.13 | 0.12 | 0.1 | 0.1 |

**Table 5** The overlap number of debris flow and landslide height and very high-class mapping units





| Landslide ⟍ Debris flow | Very low | Low | High | Very high |
|---|---|---|---|---|
| High | 3/23 | 1/23 | 5/23 | 12/23 |
| Very high | 2/26 | 2/26 | 8/26 | 11/26 |

**Table 6** Statistical parameters of the two models

| Statistical parameters ⟍ Model | Model 1 | Mode 2 |
|---|---|---|
| KMO | 0.766 | 0.643 |
| Sig. | 0.001 | 0.003 |

**Table 7** The correlation coefficients between common factors and primitive variables

| Factor | F1 | F2 | F3 |
|---|---|---|---|
| NDVI | 0.386 | -0.336 | -0.621 |
| Basin area | 0.897 | -0.007 | 0.041 |
| Main channel length | 0.984 | 0.046 | -0.023 |
| Slop angle | -0.223 | 0.829 | 0.455 |
| Maximum elevation difference | 0.744 | 0.66 | 0.011 |
| Rainfall | -0.768 | 0.33 | 0.201 |
| Average gradient of main channel | -0.753 | 0.544 | 0.106 |
| Drainage density | -0.844 | 0.06 | 0.015 |
| Roundness | 0.331 | 0.14 | 0.818 |
| Elevation | 0.133 | 0.846 | 0.382 |
| Distance to fault | -0.16 | 0.211 | 0.421 |



| | | | |
|---|---|---|---|
| Melton | -0.625 | 0.737 | 0.149 |
| Contribution rate (%) | 41.2 | 24.7 | 16.7 |
| Accumulative contribution (%) | 41.2 | 65.9 | 83.6 |

**Table 8** The correlation coefficients between common factors and primitive variables

| Factor | C1 | C2 | C3 | C4 |
|---|---|---|---|---|
| NDVI | 0.042 | -0.079 | -0.279 | -0.813 |
| Basin area | 0.802 | -0.344 | 0.057 | 0.009 |
| Main channel length | 0.885 | 0.126 | -0.196 | 0.227 |
| Slop angle | 0.009 | 0.748 | 0.58 | -0.057 |
| Maximum elevation difference | 0.801 | 0.434 | -0.128 | 0.144 |
| Rainfall | 0.197 | -0.076 | -0.487 | 0.637 |
| Average gradient of main channel | -0.744 | 0.205 | 0.15 | -0.23 |
| Drainage density | -0.776 | -0.176 | -0.267 | 0.117 |
| Roundness | -0.014 | 0.022 | 0.896 | -0.002 |
| Elevation | 0.34 | 0.746 | 0.25 | 0.326 |
| Distance to fault | 0.31 | 0.289 | -0.344 | 0.757 |
| Melton | -0.182 | 0.932 | -0.192 | 0.061 |
| Contribution rate (%) | 29.2 | 20.3 | 15.2 | 15.8 |
| Accumulative contribution (%) | 29.2 | 49.5 | 64.7 | 80.5 |

**Table 9** The overlap number of landslide and debris flow height and very-high class mapping units



| Debris flow / Landslide | Very low | Low | High | Very high |
|---|---|---|---|---|
| High | 36/167 | 33/167 | 25/167 | 43/167 |
| Very high | 48/179 | 40/179 | 25/179 | 28/179 |

**Table 10** Comprehensive evaluation of landslide-debris flow susceptibility

| Debris flow / Landslide | Low or Very low | High or Very high |
|---|---|---|
| Low or Very low | Low | Moderate |
| High or Very high | Moderate | High |

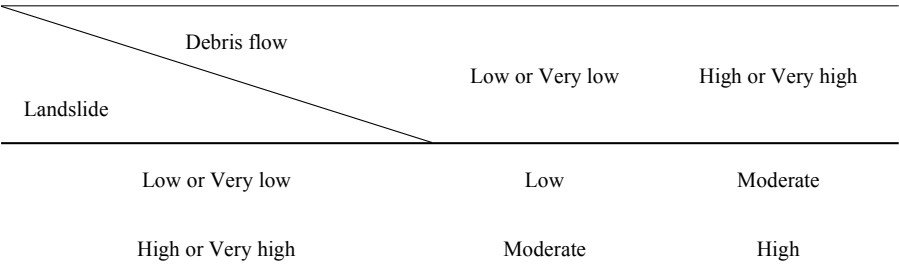

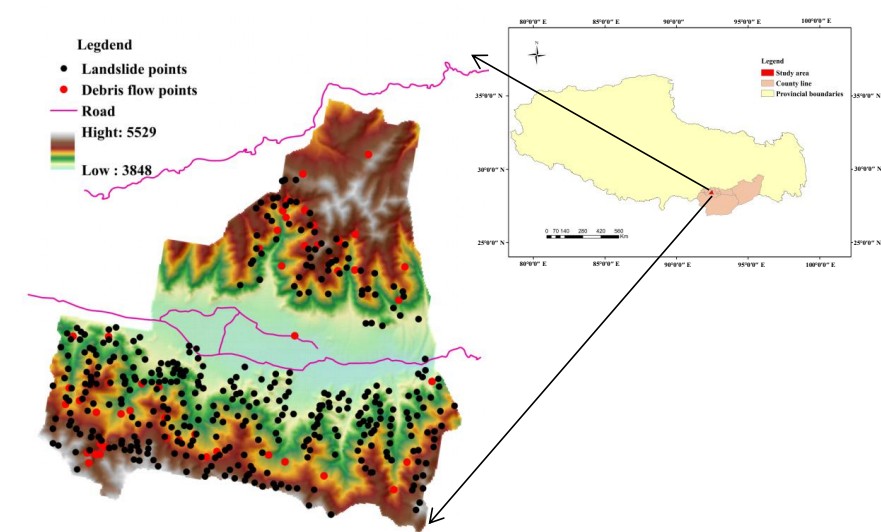

**Fig.1.** Location map of the study area showing landslide and debris flow inventory.




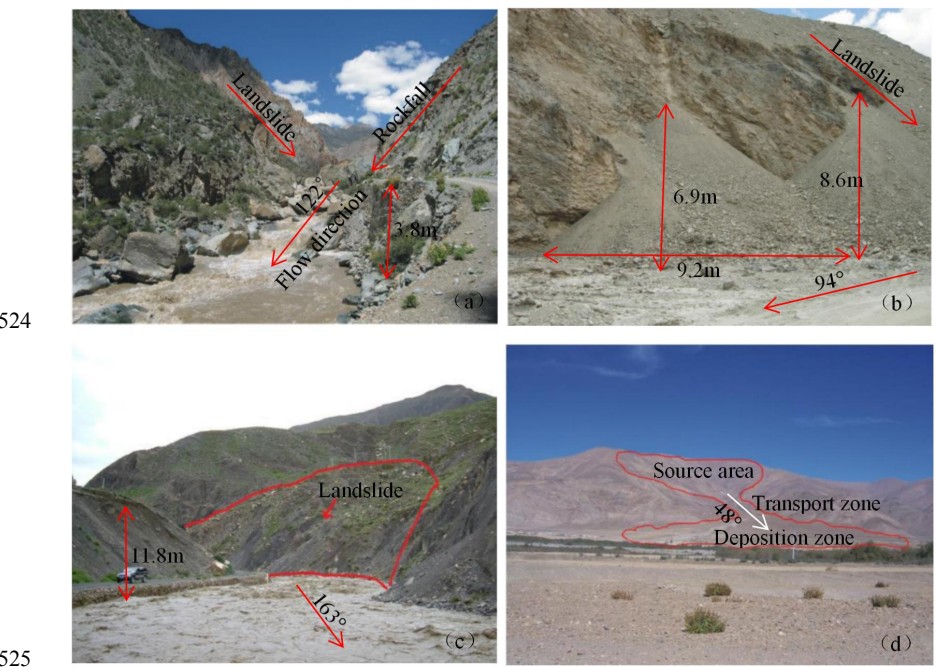

**Fig.2.** Photos of landslide or debris flow: **(a)** Lunba landslide in a tributary; **(b)** Zhenqiong landslide in
Jiayu village; **(c)** Debris flow in Misha Township; **(d)** Debris flow in Lelong Village.
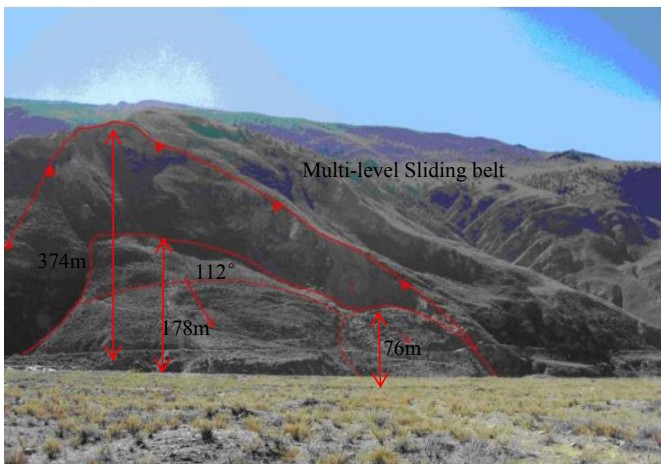
**Fig.3.** Multistage landslide in Xiongqu village


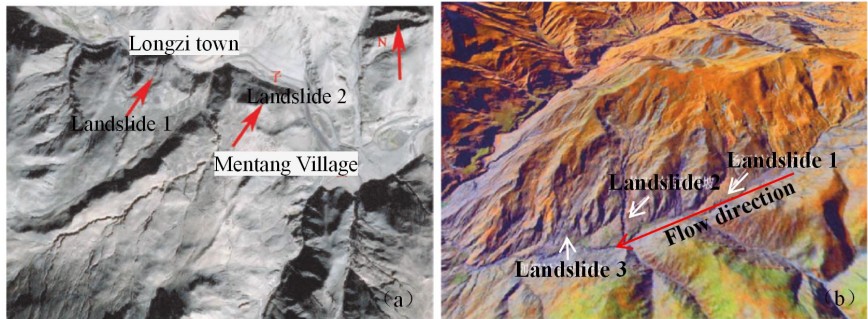


**Fig.4.** Stereo remote sensing map of landslides in Longzi Township (Tong et al., 2019): (**a**)Landslides
in Longzi town; **(b)** Landslides in Malu town.

**Fig.5.** Study area thematic maps for landslide: (**a**) Rainfall; (**b**) Profile curvature; (**c**) Maximum

elevation difference; (**d**) Average elevation; (**e**) Plan curvature; (**f**) Average slope; (**g**) Aspect;



(h)Wetness;(i)Distance to road;(j)Distance to river;(k)Distance to fault.






**Fig.6.** Study area thematic maps for debris flow:(**a**)Melton;(**b**)NDVI;(**c**)Rainfall;(**d**)Roundness;
(**e**)Maximum elevation difference;(**f**)Average elevation;(**g**)Drainage density;(**h**)Area;(**i**)
Average slope;(**j**)Average gradient of main channel;(**k**)Distance to fault.

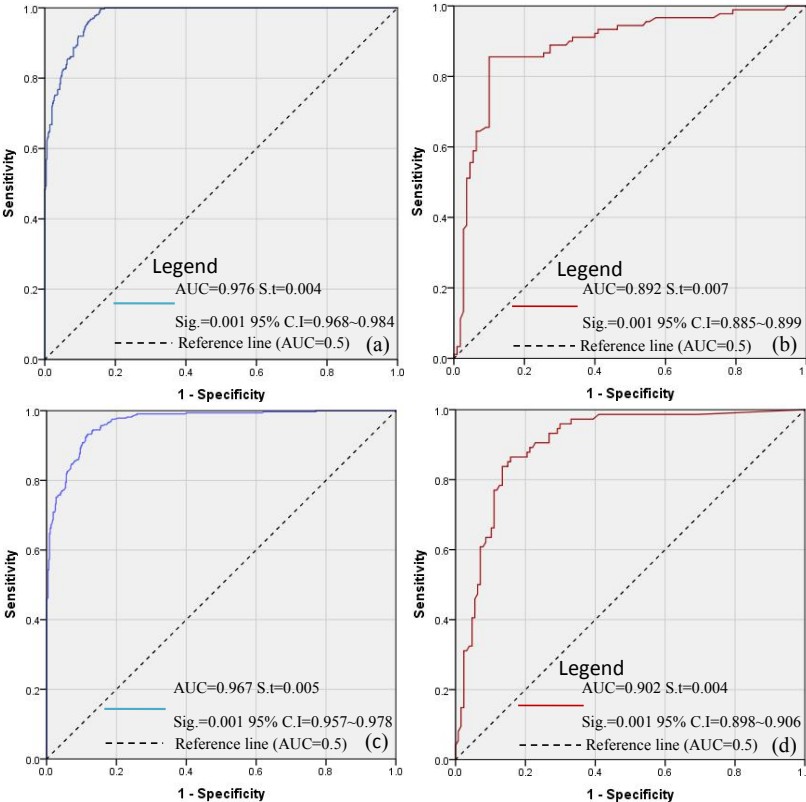


**Fig. 7.** Analysis of ROC curve for the two susceptibility maps: **(a)** Success rate curve of landslide using
the training dataset; **(b)** Prediction rate curve of landslide using the validation dataset; **(c)** Success rate
curve of debris flow using the training dataset; **(d)** Prediction rate curve of debris flow using the
validation dataset.



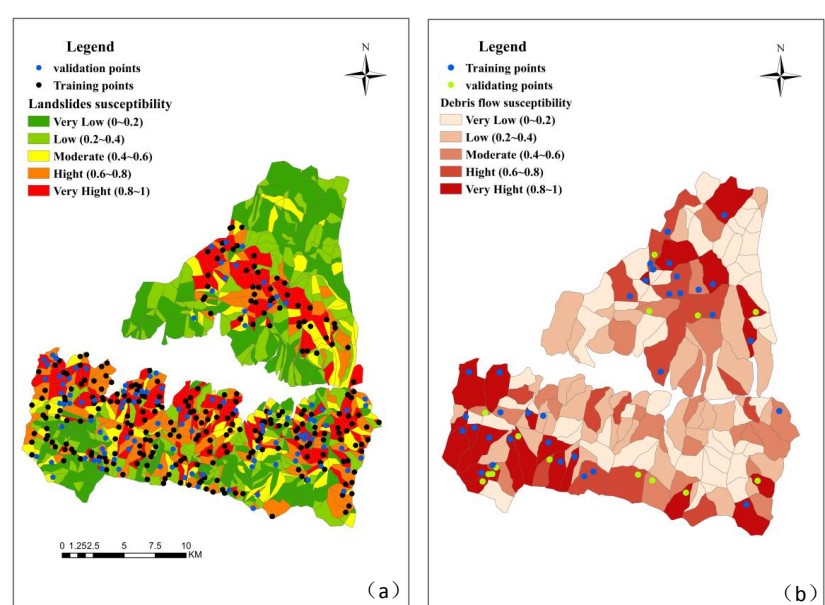

**Fig.8.** Susceptibility maps: (**a**)Landslide susceptibility zoning map; (**b**)Debris flow susceptibility

zoning map.

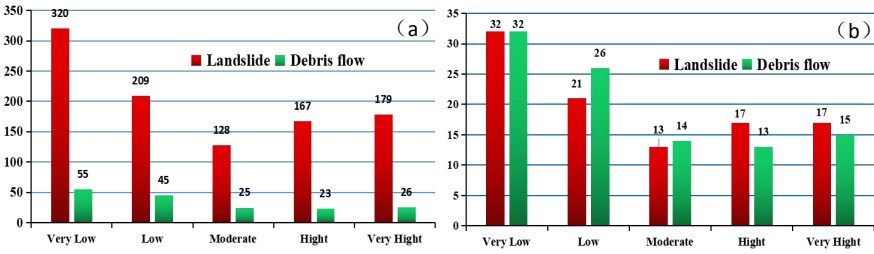

**Fig. 9.** Numbers and percentage of units in different susceptibility classes for landslide and debris flow:

**(a)** Numbers of units in different susceptibility classes for landslide and debris flow; **(b)** Percentages of

different susceptibility classes for landslide and debris flow.



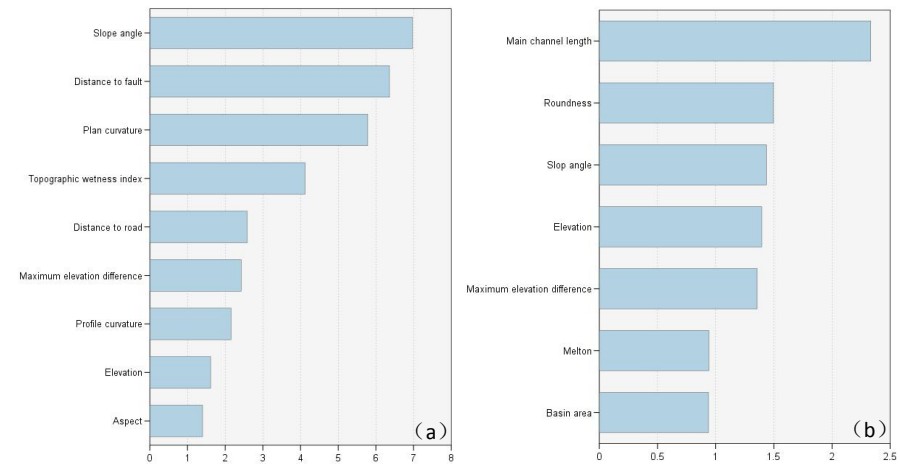

**Fig.10.** Parametric importance graphics obtained from RF model: **(a)** Parametric importance graphics

of landslide; **(b)** Parametric importance graphics of debris flow.



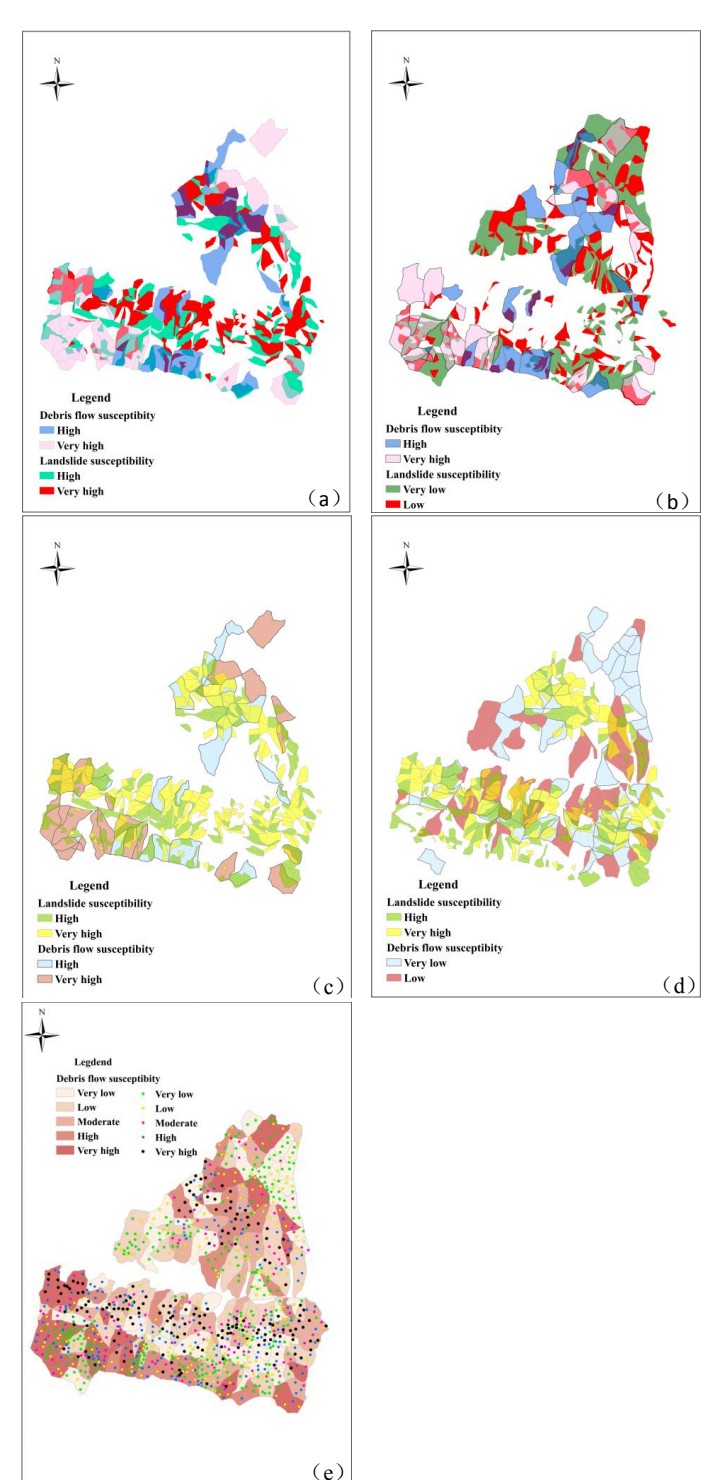





**Fig.11.** Landslide-debris flow susceptibility maps: **(a)** Height and very high-class watershed units with
high or very high slope units; **(b)** High or very high-class watershed units with low or very low slope
units; **(c)** High or very high-class slope units with high or very high-class watershed units; **(d)** Mapping
units.


