# Peer review of "Exploring the potential relationship between the occurrence"

_Natural Hazards and Earth System Sciences, 2020_

## Referee Comment (RC1) · Anonymous Referee #1 · 28 Nov 2020

General Comments

In this paper, landslide susceptibility analysis and debris flow susceptibility analysis is carried out using Random Forests over the same area and the two resultant hazard maps are compared. This is something that is not usually done in ground failure hazard assessments and I find the conclusions from this paper are interesting. The study was designed well and the figures are good, but in some places the work is not explained clearly enough or more information would help the reader to understand.

One thing I would also like to know is this: how have you chosen the training and test data for the study? Are they randomly selected in time and space? Or are historical landslides and debris flows being used to predict the locations of recent landslides and debris flows? I am also confused by the decision made by the authors to convert

all their continuous input factors (e.g. aspect) into categorical variables, as Random Forests work well with continuous variables. This is not something I have seen done in other studies using RF for landslide susceptibility mapping. If there is a specific reason that the authors have chosen to do this, it should be explained in Section 2.4. If this is the case, how did the authors choose the number of categories for each input factor?

Specific Comments

Line 88: I would also like to know if the landslides are mapped as points or polygons in your dataset (I assume points since they are shown as points in Figure 1) and the mapping resolution, or at least the resolution of the google earth images used in the landslide mapping. It would be useful to know what proportion of landslides were mapped using the different methods (i.e. historical records versus google earth image interpretation) and how far back in time your historical records go.

Line 101: What is meant by "There is no physical relationship between a grid-cell and slope" – do you mean that slope will vary within a grid cell?

Section 2.4: In this section, I think more justification is needed for the choice of controlling factors. I would also divide the section into "factors used in landslide susceptibility assessment", "factors used in debris flow susceptibility assessments" and "factors used in both". I think you have done this, but I would make it clear at the beginning of each paragraph which input factors you are describing.

Line 113: It's true that different parameters are used in soil slide and debris flow susceptibility assessment. However, there is also quite a lot of difference between the factors used by different landslide susceptibility assessments.

Line 120: When you say NDVI, are you using pre-event NDVI, as a proxy for land cover type, or post-event NDVI as a direct measurement of vegetation removal caused by the debris flow?

Lines 148-153: What are the sources of your datasets? (the geological map, the DEM,

the roads, the faults, the rainfall)

Line 149: What is the source of your DEM data?

Line 187: There are several options for optimisation in sci-kit learn. Which one did you use?

Line 193: I think here you mean you are analysing the relative importances of the conditioning factors.

Line 201: Specify here that AUC of 0.5 = No Skill for ROC curve, otherwise people might think the scale is from 0-1

Line 232: When you say "disaster points", you are referring to debris flows, so I would just say observed debris flows.

Line 257 I don't understand what is meant here by "factor analysis". What exactly has been done?

Line 260: Please explain how KMO testing works and how to interpret the values

Line 263-265: When you say model 1 and model 2, are these the landslide SZM and debris flow SZM models respectively?

Line 312: You should give examples of studies that use logistic regression and discriminant analysis here to back up your statement

Line 314: Random Forests have been applied to landslide susceptibility in several previously published works, which should be referenced here. Some examples:

Chen, W., Xie, X., Wang, J., Pradhan, B., Hong, H., Bui, D.T., Duan, Z. and Ma, J., 2017. A comparative study of logistic model tree, random forest, and classification and regression tree models for spatial prediction of landslide susceptibility. Catena, 151, pp.147-160.

Catani, F., Lagomarsino, D., Segoni, S. and Tofani, V., 2013. Landslide susceptibility

estimation by random forests technique: sensitivity and scaling issues. Natural Hazards and Earth System Sciences, 13(11), p.2815.

Zhang, K., Wu, X., Niu, R., Yang, K. and Zhao, L., 2017. The assessment of landslide susceptibility mapping using random forest and decision tree methods in the Three Gorges Reservoir area, China. Environmental Earth Sciences, 76(11), pp.1-20.

Section 5.2: If I understand correctly, what you are saying here is that landslide susceptibility maps should not be used in debris flow hazard assessment and vice versa. This seems to me to be an important conclusion from this paper and should be stated more clearly.

Line 343: this is not very clear can you give a more specific example?

Table 1: The layout of this table is a bit strange, having a single row with so much information in it. I also think the parameters may not make any sense to someone who has not used the sci-kit learn package for example "max_features, sqrt".

Table 1: You have two models here: one for landslides and one for debris flows. Did the optimisation technique you used yield the same optimum parameters for both models?

Figure 10: There is no label for the X axis. I assume it should be "importance (%)"

Technical Corrections

Line 52: There is no space between flow"(Varnes,

Line 94: There is no space between 7.1(Fig.4)

Line 148: Do you mean the "raw data" rather than the "row data"?

Line 166: "curves( Green" the space should be before the bracket

Lines 170-172: Do these need to be separately numbered equations? Also in "Sensitivity" and "Accuracy" some of the word is in italics and some is not.

Line 189: Are there two spaces between "trees" and "and"?

Line 260: significance (Sig) was defined earlier in the manuscript

Line 337: No space between "respectively(Fig.10)"

Line 286: Zonation not Zoination
* * *

---

## Referee Comment (RC2) · Anonymous Referee #2 · 16 Dec 2020

The paper is suitable for NHESS. Unfortunately is not acceptable in present form. There are the following main deficiencies:

1) Introduction should partially enlarged for a better and specific presentation of debris-flow phenomena (see below). 2) Section 3 is not clear: authors should explain the method or model they used. There is no connection between 3.1 and 3.2. Moreover, the presentation of the RF model is not clear. 3) Section 4. The procedure should be introduced at the beginning: at first evaluation of the training data set and after that of the remaining data set. Moreover, there is some confusion on the presentation of data analysis (e.g. see the comment to lines 252-253). 4) Section 5. All the assumptions claimed by the authors should be supported by results shown at the previous section. Otherwise, all this section is thin air. In other words, each assumption should be justi-

fied by the findings of the previous section. 5) English form is not always acceptable

Introduction

The writer suggests a brief characterization of debris flows based on the triggering mechanisms and conditions to explain the phenomenon and avoid confusion, as that below: Most of debris flows are runoff generated (Imaizumi et al. 2006; Coe et al., 2008, Gregoretti & Dalla Fontana, Ma et al., 2018). In many cases they occur on a channel bed for the entrainment into abundant runoff of debris supplied by deep or shallow slides of slopes incised by the channel. (Theule et al. (2012), Hurlimann et al. (2014), Imaizumi et al. 2019, Zhou et al.,. 2019; Simoni et al., 2020). Conversely, landslide or natural dam failure that evolve into a debris flow (Iverson et al., ; Kean et al., 2013) are not frequent. Moreover, it is not clear the way the potential relationship between debris flow and landslide is approached through the separated susceptibility analysis: some concise information could help the reader.

Other spotted errors and comments are as follows:

Line 75 perhaps "surface" instead of "area"

Line 82 "Three types of lithology were mainly observed" rather than "There were three common lithology observed"

Line 85 "Main common disasters in the study area mainly consist" instead of "The disasters in the study area mainly consist"

Lines 104-105 "The geometry of debris flow is better represented by a polygon or a set of polygons in vector format" Which is the sense of such a sentence? Authors should explain, as in the case of landslide which typology of unit is preferable for debris flow.

Line 115: Moreover, before availability

Lines 114-116 "The occurrence of debris flow emphasizes the indispensability of provenience, topography and triggering factors. Availability, reliability, and practicality of the

factor data were also considered (van Westen et al., 2008)." Such period should be postponed to that at lines 116-119.

Line 122: elevation of what?

Line 124: Figure 5 concerns landslide. The figure 6 concerning debris flow should be also added. The writer suggests to distinguish the reference to these two figures by means of the phenomenon.

Line 126 please separate the controlling factors concerning landslide from those concerning debris flows.

Line 127 Why do the authors use reclassify and not classify?

Line 128: the proximity of roads, rivers rather than roads, river....Therefore the distance from roads, river was classified ..............

Lines 137-138 "Considering the correlation between the two controlling factors, basin area and main channel length are represented by the same graph, which was reclassified into four classes (Fig.6h)." unclear sentence.

Line 148 "Totally 18 factors are obtained by processing the row data in the ArcGIS 10.2 platform." Perhaps the values of 18 controlling factors were classified by processing......

Line 149 The DEM size, 30 m seems too large. The author should justify it. Please consider that Boreggio et al. (2018) suggested the use of 1 m grid size.

Line 153 "under study as a reference." Unclear expression.

Lines 157-158 "data set" rather than "set"

Lines 159-161 The partition of landslide inventory is approached .......... among them that of one time random selection () is the most used.

Lines 179-180 "RF uses the bagging technique (bootstrap aggregation) to select, at

each node of the tree, random samples of variables and observations as the training data set for model calibration." Unclear sentence

Lines 186-194. Unclear period.

Line 197 "training data set" rather than "training set"

Line 198 "of for"????

Line 199 " with a sensitivity value"

Line 203 "for models" ??????

Line 206 Values of 88.69 and 86.05 % are claimed for sensitivity and specificity respectively. Why at the previous lines the values are 91.62 and 89.96%? Please explain

Line 207 "with a value"

Line 208 "training model"???? perhaps it is training dataset

Line 214 delete "reached 179"

Lines 215, 216, 231 what does relate the percentage? The total number of units? Please specify

Line 219 "were" instead of "was"

Line 230 delete "reached to 26"

Line 235 which is the sense of the following sentence? "which has significant influence on the occurrence of debris flow"

Line 236 substitute "which are" with ":"

Figure 8. The writer suggests the use of the same colours for both the susceptibility maps

Lines 249-251 "There are 23 watershed units belonging to high-class in the debris

flow susceptibility zoning map (Fig.8), of which 17 units are covered with high or very high-class slope units in the landslide zoning map (Table 5)." 1) it is better substitute "are covered" with "correspond to high or very high-class" Moreover, add the following sentence: "Therefore, there are 6 units that does not overlap (about 26%)."

Line 251 about the 4 watershed units: do the belong to the 6 watershed units with no high or very-high slope units?

Lines 252-253 which susceptibility maps do belong the 19 high and very-high class watershed units (19)

Line 269 which two models?

Lines 313-314 "has been little used until now for susceptibility analysis of landslide and debris flows " instead of "has less application in landslide and debris flow analysis"

Line 322 "from the concept" unclear expression

Lines 326-329 this period should be summarized in a more concise and clear form

Line 338 Where the relationship between landslide and debris flow is illustrated?

Lines 348-349 "The fact that the appropriate prediction method and mapping units applied to the two disasters makes it possible to merge the two zoning maps" Which appropriate prediction method? Which is sense of this sentence?

Line 359 "models based on random forest " if the authors mean "based on RF models" the expression is unclear (see lines 197; 241). This ambiguity is elsewhere present in the submitted manuscript.

Points 2 and 3 of the conclusion could be merged in a unique one. This point should begin after explaining that there is no potential relationship between the occurrence of the two considered phenomena. After that, the authors could explain the reasons in 2.1. and 2.2 corresponding to the points 2 and 3 of the submitted work.

[Figure]

REFERENCES

Boreggio, M., Bernard, M. Gregoretti C. (2018) Evaluating the influence of gridding techniques for Digital Elevation Models generation on the debris flow routing modelling: A case study from Rovina di Cancia basin (North-eastern Italian Alps). Frontier in Earth Sciences, doi: 10.3389/feart.2018.00089

Coe, J. A., Kinner, D. A., and Godt, J. W. (2008). Initiation conditions for debris flows generated by runoff at Chalk Cliffs, Central Colorado. Geomorphology 96, 270–297. doi: 10.1016/j.geomorph.2007.03.017

Gregoretti, C., and Dalla Fontana, G. (2008). The triggering of debris fl ow due to channel ‐ bed failure in some alpine headwater basins of the Dolomites: Analyses of critical runoff. Hydrological Processes, 22, 2248 – 2263.

Hurlimann M., Abanco C., Moya, J., Vilajosana I. 2013. Results and experiences gathered at the Rebaixader debris-flow monitoring site, Central Pyrenees, Spain. Landslides. doi:10.1007/s10346-013-0452-y 161-175

Imaizumi F, Sidle RC, Tsuchiya S, Ohsaka O. 2006. Hydrogeomorphic processes in a steep debris flow initiation zone. Geophysical Research Letters 33: L10404.

Imaizumi F, Masui T, Yokota Y, Tsunetaka H, Hayakawa YS, Hotta N. 2019. Initiation and runout characteristics of debris flow surges in Ohya landslide scar, Japan. Geomorphology 339: 58 – 69.

Iverson RM, Reid ME, Lahusen RG. 1997. Debris-flow mobilization from landslides. Annual Review of Earth and Planetary Sciences 25: 85–136.

Kean J.W., McCoy S.W., Tucker G.E., Staley D.M., Coe J.A. (2013). Runoff-generated debris flows: Observations and modeling of surge initiation, magnitude, and frequency. Journal of Geophysical Research Earth Surface, VOL. 118, 2190–2207. doi:10.1002/jgrf.20148, 2013

Ma C., Deng J. Wang R. (2018) Analysis of the triggering conditions and erosion of a runoff triggered debris flow in Miyun County, Beijing, China. Landslide, DOI 10.1007/s10346-018-1080-3

Theule, J.I., Liebault, F., Loye, A., Laigle, D., and Jaboyedoff, M., 2012. Sediment budget monitoring of debris flow and bedload transport in the Manival Torrent, SE France.

Zhou, W., Fan. J., Tang, C., Yang, G. (2019) Empirical relationships for the estimation of debris flow runout distances on depositional fans in the Wenchuan earthquake zone. Journal of Hydrology, 577, https://doi.org/10.1016/j.jhydrol.2019.123932

---

## Author Comment (AC2) · 17 Dec 2020

Dear Editors and Reviewers: Thank you for your letter and for the Reviewers' comments concerning our manuscript entitled "Exploring the potential relationship between the occurrence of debris flow and landslide" (ID: NHESS-294). Those comments are all valuable and very helpful for revising and improving our paper, as well as the important guiding significance to our researches. We have studied comments carefully and have made correction which we hope meet with approval. Revised portion are marked in red in the paper. The main corrections in the paper and the responds to the Reviewer's comments are as flowing:

1)Introduction should partially enlarged for a better and specific presentation of debris

flow phenomena (seebelow). Response: We have modified it according to the comments. Line 41, 44-45, 48-49.

2)Section3 is not clear:authors should explain the method or model they used. There is no connection between 3.1 and 3.2. Moreover, the presentation of the RF model is not clear. Response: We have re-write related information of the modeling of RF. Section 3.1 and 3.2 belong to the method used in this study. Section 3.1 explain the sampling strategy and elevation indexes for RF models. Section 3.2 introduce RF model.

3)Section4.The procedure should be introduced at the beginning:at first evaluation of the training dataset and after that of the remaining dataset. Moreover,there is some confusion on the presentation of data analysis(e.g.see the comment to lines252-253). Response: Yes, we have introduced the performance of RF model in terms of training data set first and then compared the results with validation data set. Section 4.3 is confusing because we have to compare the results of debris flow and landslide. The maps were both reclassified into five levels and we tried to present them on the same map. We have checked the results and make it easier to be understood.

4)Section5. All the assumptions claimed by the authors should be supported by results shown at the previous section. Otherwise, all this section is thin air. In other words,each assumption should be justified by the findings of the previous section. Response: Yes, we could not agree more. The results we obtained indicate that RF was suitable for landslide susceptibility mapping, there is no determined relationship between debris flow and landslide, it is feasible to map two kinds of disaster in the same susceptibility map.

5) English form is not always acceptable Response: We have checked the whole manuscript again.

5)Introduction The writer suggests a brief characterization of debris flows based on the triggering mechanisms and conditions to explain the phenomenon and avoid confusion,as that below:Most of debris flows are runoff generated (Imaizumietal.2006;Coeetal.,2008,Gregoretti&DallaFontana,Ma etal.,2018).In many cases they occur on a channel bed for the entrainment into abundant runoff of debris supplied by deep or shallow slides of slopes incised by the channel. (Theulee-tal.(2012),Hurlimannetal.(2014),Imaizumietal.2019,Zhouetal.,.2019;Simonietal.,2020). Conversely,landslide or natural dam failure that evolve into a debris flow(Iversonetal.,;Keanetal.,2013) are not frequent. Moreover,it is not clear the way the potential relationship between debris flow and landslide is approached through the separated susceptibility analysis:some concise information could help the reader. Response: We have add related information in the Introduction.

Other spotted errors and comments are as follows: Line75 perhaps "surface"instead of "area" Response:We modified it accordingly. Line78

Line 82 "Three types of lithology were mainly observed"rather than"There were three common lithology observed" Response:We modified it accordingly. Line85

Line85 "Main common disasters in the study area mainly consist"instead of"The disas-ters in the study area mainly consist" Response:We have already modified it. Line88 Lines104-105"The geometry of debris flow is better represented by a polygon or a set of polygons in vector format"Which is the sense of such a sentence?Authors should explain,as in the case of landslide which typology of unit is preferable for debris flow. Response:We have added related information. The watershed unit is preferable for de-bris flow.line 109-110. Line115:Moreover, before availability Response:We have added related information. Lines114-116"The occurrence of debris flow emphasizes the indis-pensability of provenience, topography and triggering factors.Availability,reliability,and practicality of thefactor data were also considered (vanWestenetal.,2008)."Such period should be postponed to that at lines116-119. Response:We have already modified it.

Line122:elevation of what? Response:Maximum elevation difference is another condi-tioning factor.ine124:Figure 5 concerns landslide. The figure 6 concerning debris flow should be also added.The writer suggests to distinguish the reference to these two

figures by means of the phenomenon. Response:We have already modified it. line126 please separate the controlling factors concerning landslide from those concerning debris flows. Response:We have already modified it.

Line127 Why do the authors use reclassify and not classify? Response:Reclassify is an tool in ArcGIS platform. Line128: the proximity of roads,rivers rather than roads,river....Therefore the distance from roads,river was classified

Lines137-138"Considering the correlation between the two controlling factors,basin area and main channel length are represented by the same graph,which was reclassified into four classes(Fig.6h)."unclear sentence. Response:We have already modified it. Line 148-149

Line148"Totally 18 factors are obtained by processing the row data in the ArcGIS10.2 platform."Perhaps the values of 18 controlling factors were classified by processing...... Response: We have already modified it. Line 161-162

Line149 The DEM size,30 m seems too large.The author should justify it.Please consider that Boreggioetal.(2018) suggested the use of 1m grid size. Response:The DEM size is accessible for 30m and 90m. Some studies use 5m or 1m by resampling tool of ArcGIS. However, 30m is the most common.

Line153"under study as a reference."Unclear expression. Response:We have already modified it.

Lines157-158"data set"rather than"set" Response:We have already modified it. Line 172

Lines159-161The partition of landslide inventory is approached..........among them that of one time random selection() is the most used. Response:We have already modified it. Line 175-177

Lines179-180"RF uses the bagging technique (bootstrap aggregation)to select, at each node of the tree,random samples of variables and observations as the training data set

for model calibration."Unclear sentence Response:We have already modified it. Line 194-192

Lines186-194.Unclear period. Response:We have already modified it. Line 200-202.

Line197"training data set"rather than"training set" Response:We have already modified it.

Line198"of for"???? Response:We have already modified it.

Line199"with a sensitivity value" Response:We have already modified it.

Line203"for models"?????? Response:We have already modified it.

Line206 Values of 88.69 and 86.05% are claimed for sensitivity and specificity respectively.Why at the previous lines the values are91.62 and 89.96%? Please explain Response:The data set were divided into two groups, one for training, the other for validation. Therefore, the sensitivity and specificity values were different.

Line207"with a value" Response:We have already modified it.

Line208"training model"???? perhaps it is training dataset Response:We have already modified it.

Line214 delete"reached 179" Response:We have already modified it.

Lines215,216,231 what does relate the percentage?The total number of units?Please specify Response:We have already modified it.

Line219"were" in stead of "was" Response:We have already modified it.

Line230 delete"reached to 26" Response:We have already modified it.

Line235 which is the sense of the following sentence¿'which has significant influence on the occurrence of debris flow Response:We have already modified it.

Line236 substitute "which are"with":" Response:We have already modified it.

Figure8.The writer suggests the use of the same colours for both the susceptibility maps Response: We have compared the results before and found it better when the maps were made from different colour to highlight the difference between debris flow and landslide.

Lines249-251"There are 23 watershed units belonging to high-class in the debris flow susceptibility zoning map (Fig.8),of which17 units are covered with high or very high-class slope units in the landslide zoning map(Table5)."it is better substitute"are covered"with"correspond to high or very high-class"Moreover, add the following sentence:"Therefore,there are 6 units that does not overlap (about26%)." Response:We have already modified it.

Line251 about the 4 watershed units:do the belong to the 6 watershed units with no high or very-high slope units? Response: The susceptibility maps were reclassified into five levels as very low, low, moderate, high and very high.

Lines252-253 which susceptibility maps do belong the 19 high and very-high class watershed units(19) Response:Watershed units were for debris flow and slope units were for landslide.

Line269 which two models? Response: We have added detail information.

Lines 313-314 "has been little used until now for susceptibility analysis of landslide and debris flows"instead of "has less application in landslide and debris flow analysis" Response:We have already modified it.

Line322" from the concept"unclear expression Response:We have already modified it.

Lines326-329 this period should be summarized in a more concise and clear form Response:We have already modified it.

Line 338 Where the relationship between landslide and debris flow is illustrated? Response:We have added related information.

Lines 348-349 "The fact that the appropriate prediction method and mapping units applied to the two disasters makes it possible to merge the two zoning maps"Which appropriate prediction method?Which is sense of this sentence? Response: Random forest has proved its superiority in this study. Mapping two kinds of disaster in the same map has not been explored before and we try to explain why and how does it works.

Line 359 "models based on random forest"if the authors mean"based on RF models"the expression is unclear(seelines197;241).This ambiguity is elsewhere present in the submitted manuscript. Response:We have already modified it.

Points 2 and 3 of the conclusion could be merged in a unique one.This point should begin after explaining that there is no potential relationship between the occurrence of the two considered phenomena. After that,the authors could explain the reasons in 2.1.and2.2 corresponding to the points 2 and 3 of the submitted work. Response: We have modified it.

Finally, we have added related reference based on the comments.

We appreciate for Editors and Reviews' warm work earnestly, and hope that the correction will meet with approval. Once again, thank you very much for your comments and suggestions. With best regard, Yours sincerely, Zhu Liang Jilin University

---

## Author Response (AR1)

Dear Editors and Reviewers

Thank you for your letter and for the Reviewers' comments concerning our manuscript entitled "Exploring the potential relationship between the occurrence of debris flow and landslide" (ID: NHESS-294). Those comments are all valuable and very helpful for revising and improving our paper, as well as the important guiding significance to our researches. We have studied comments carefully and have made correction which we hope meet with approval. Revised portion are marked in red in the paper. The main corrections in the paper and the responds to the Reviewer's comments are as flowing:

Response to the first Reviewer:

General Comments In this paper, landslide susceptibility analysis and debris flow susceptibility analysis is carried out using Random Forests over the same area and the two resultant hazard maps are compared. This is something that is not usually done in ground failure hazard assessments and I find the conclusions from this paper are interesting. The study was designed well and the figures are good, but in some places the work is not explained clearly enough or more information would help the reader to understand.

Response: Thank you for your approval of our work. It is true that there is few manuscripts related in ground failure hazard assessments and we will try our best to make the idea more clear.

One thing I would also like to know is this: how have you chosen the training and test data for the study? Are they randomly selected in time and space? Or are historical landslides and debris flows being used to predict the locations of recent landslides and debris flows?

Response: In second 3.1, we choose the 5-fold cross validation procedure. The data consists of negative and positive samples. Conditioning that the number of samples are limited and we selected the all the samples in time during 1970～2010 for modeling. Related information have been added on line 96, 98-100.

I am also confused by the decision made by the authors to convert all their continuous input factors (e.g. aspect) into categorical variables, as Random Forests work well with continuous variables. This is not something I have seen done in other studies using RF for landslide susceptibility mapping. If there is a specific reason that the authors have chosen to do this, it should be explained in Section 2.4. If this is the case, how did the authors choose the number of categories for each input factor?

Response: It is true that random Forests work well with continuous variables. We did not convert the continuous input factors into categorical variables. We aim to reclassify variables as the thematic map will become more concise. As for the number of categories, we referred to related references which have been published.

S.Chen, Z. Miao, L. Wu and Y. He, "Application of an Incomplete Landslide Inventory and One Class Classifier to Earthquake Induced Landslide Susceptibility Mapping," in a IEEE Journal of Selected Topics in Applied Earth Observations and Remote Sensing, vol. 13, pp. 1649-1660, 2020, doi: 10.1109/JSTARS.2020.2985088.

Zhu Liang, Wang Changming and Kaleem-Ullah-Jan Khan. Application and comparison of different ensemble learning machines combining with a novel sampling strategy for shallow

landslide susceptibility mapping. Stoch Environ Res Risk Assess (2020c). https://doi.org/10.1007/s00477-020-01893-y

Specifific Comments Line 88: I would also like to know if the landslides are mapped as points or polygons in your dataset (I assume points since they are shown as points in Figure 1) and the mapping resolution, or at least the resolution of the google earth images used in the landslide mapping. It would be useful to know what proportion of landslides were mapped using the different methods (i.e. historical records versus google earth image interpretation) and how far back in time your historical records go.

Response: We are agreed with the comment. The landslide locations are recorded as a point which are showed in Fig.1. We have provided detail information like time about records (line 96, 99-100).

Line 101: What is meant by "There is no physical relationship between a grid-cell and slope" – do you mean that slope will vary within a grid cell?

Response: Landslides are the result of slope processes acting at different spatial and temporal scales that result in geomorphological forms of very different shapes and sizes that are difffificult to capture by grid-cells accurately. The geometry of a landslide is better represented by a polygon or a set of polygons in vector format; unless the size of the grid-cell is very small compared to the size of the landslide.We have referred to the original manuscript and make it more clear (line106-107, 109-110).

Section 2.4: In this section, I think more justification is needed for the choice of controlling factors. I would also divide the section into "factors used in landslide susceptibility assessment", "factors used in debris flow susceptibility assessments" and "factors used in both". I think you have done this, but I would make it clear at the beginning of each paragraph which input factors you are describing.

Response: We are agreed with the comment. We have added related information and divide the sections clearly (line 122-124, 131-141).

Line 113: It's true that different parameters are used in soil slide and debris flow susceptibility assessment. However, there is also quite a lot of difference between the factors used by different landslide susceptibility assessments.
Response: We are agreed with the new expression which is more accurate and clear (line 118-119).

Line 120: When you say NDVI, are you using pre-event NDVI, as a proxy for land cover type, or post-event NDVI as a direct measurement of vegetation removal caused by the debris flow?
Response: NDVI reflects the vegetation conditions in the area and we use the preevent NDVI as a proxy for land cover type.

Lines 148-153: What are the sources of your datasets? (the geological map, the DEM, the roads, the faults, the rainfall)
Response: We are agreed with the comment. We have provided related information on line 162-168.

Line 149: What is the source of your DEM data?
Response: We provide related link where we download the DEM data on line 162-168.

Line 187: There are several options for optimisation in sci-kit learn. Which one did you use?
Response: Cross-Validation were applied in our work.

Line 193: I think here you mean you are analysing the relative importances of the conditioning factors.
Response: Yes, what you think is correct and we have explain it correctly on line 206.

Line 201: Specify here that AUC of 0.5 = No Skill for ROC curve, otherwise people might think the scale is from 0-1.

Response: We are agreed with the comment. We have explain the AUC value more clearly on line 212-213.

Line 232: When you say "disaster points", you are referring to debris flows, so I would just say observed debris flows.
Response: Disaster points in our work referred to all landslide locations not just to debris flows.

Line 257 I don't understand what is meant here by "factor analysis". What exactly has been done?
Response: Factor analysis is a method which is usually used for dimensionality reduction and exploring the major factors. We have added related information on line 272-278.

Line 260: Please explain how KMO testing works and how to interpret the values.
Response: We have added related information on line 275-278.

Line 263-265: When you say model 1 and model 2, are these the landslide SZM and debris flow SZM models respectively?
Response: 36 watershed units with distribution of high or very high-grade slope units were taken as model 1 and the left 8 watershed units as model 2, which has been explained on line 262-264. The models established is to explore the major conditioning factors for analyzing the reason why some high or very high-grade susceptibility watershed units are covered with low susceptibility slope units.

Line 312: You should give examples of studies that use logistic regression and discriminant analysis here to back up your statement
Response: We are agreed with the comment. We have added related references here (line 329 and 333).

Line 314: Random Forests have been applied to landslide susceptibility in several previously published works, which should be referenced here. Some examples: Chen, W., Xie, X., Wang, J., Pradhan, B., Hong, H., Bui, D.T., Duan, Z. and Ma, J., 2017. A comparative study of logistic model tree, random forest, and classification and regression tree models for spatial prediction of landslide susceptibility. Catena, 151,pp.147-160.
Catani, F., Lagomarsino, D., Segoni, S. and Tofani, V., 2013. Landslide susceptibility estimation by random forests technique: sensitivity and scaling issues. Natural Hazards and Earth System Sciences, 13(11), p.2815. Zhang, K., Wu, X., Niu, R., Yang, K. and Zhao, L., 2017. The assessment of landslide susceptibility mapping using random forest and decision tree methods in the Three Gorges Reservoir area, China. Environmental Earth Sciences, 76(11), pp.1-20.
Response: We are agreed with the comment. We have added related references here (line332).

Section 5.2: If I understand correctly, what you are saying here is that landslide susceptibility maps should not be used in debris flow hazard assessment and viceversa. This seems to me to be an important conclusion from this paper and should be stated more clearly.

Response: Yes, it is correct that different kinds of landslides should be evaluated respectively conditioning that conditioning factors and scale varies. We add related information on line 355-356.

Line 343: this is not very clear can you give a more specifific example?
Response: We adjusted the language order and enhanced the before-and-after logic (360-363).

Table 1: The layout of this table is a bit strange, having a single row with so much information in it. I also think the parameters may not make any sense to someone who has not used the sci-kit learn package for example "max_features, sqrt".
Response: We are agreed with the comment. The application of machine learning usually involves several hyper-parameter needed to be tuned before modeling. Different machine learning methods need different optimum parameters or different optimisation technique will generate different values of parameters which are not easy to implement and even means nothing to someone who are not skilled at modeling. The main aim of our work is to explore the relationship between soil slides and debris flow by mapping respectively not for the comparison of different methods. Therefore, it will be easier to be understood if Table 1 is removed. And we decided to removed Table 1 and related information.

Table 1: You have two models here: one for landslides and one for debris flflows. Did the optimisation technique you used yield the same optimum parameters for both models?
Response: The optimum parameters are not the same because the number of samples of landslide and debris flow are different. On the other hand, the conditioning factors for landslide and debris flow are also different.

Technical Corrections
Line 52: There is no space between flow"(Varnes, Response:We have make it correct.
Response:We have make it correct.

Line 94: There is no space between 7.1(Fig.4) Response:We have make it correct.
Response:We have make it correct.

Line 148: Do you mean the "raw data" rather than the "row data"? Response: It should be raw data. Line 166: "curves( Green" the space should be before the bracket
Response: We have make it correct.

Lines 170-172: Do these need to be separately numbered equations? Also in "Sensitivity" and "Accuracy" some of the word is in italics and some is not.
Response: Accuracy, Sensitivity and specificity are three similar indexes for evaluating the performance of model and we have referred to some other papers that the equations are listed in together. TP, TN, FN and FP should be in italics and we have made them correct.

Line 189: Are there two spaces between "trees" and "and"?
Response: We have make it correct.

Line 260: signified significance (Sig) was defifined earlier in the manuscript

Response: We have make it correct.

Line 337: No space between "respectively(Fig.10)"

Response: We have make it correct.

Line 286: Zonation not Zoination

Response: We have make the word correct.

**Response to Reviewer 2#:**

1) Introduction should partially enlarged for a better and specifific presentation of debris flow phenomena (see below).

Response: We have modified it according to the comments. Line 41, 44-45, 48-49.

2) Section3 is not clear:authors should explain the method or model they used. There is no connection between 3.1 and 3.2. Moreover, the presentation of the RF model is not clear.

Response: We have re-write related information of the modeling of RF(line194-195, 200-202, 205-206). Section3.1 and 3.2 belong to the method used in this study. Section 3.1 explain the sampling strategy and elevation indexes for RF models. Section 3.2 introduce RF model.

3) Section4.The procedure should be introduced at the beginning:at first evaluation of the training dataset and after that of the remaining dataset. Moreover,there is some confusion on the presentation of data analysis (e.g.see the comment to lines252-253).

Response: Yes, we have introduced the performance of RF model in terms of training data set first and then compared the results with validation data set. Section 4.3 is confusing because we have to compare the results of debris flow and landslide. The maps were both reclassified into five

levels and we tried to present them on the same map. We have checked the results and make it easier to be understood.

4) Section5. All the assumptions claimed by the authors should be supported by results shown at the previous section. Otherwise, all this section is thin air. In other words,each assumption should be justified by the findings of the previous section.

Response: Yes, we could not agree more. The results we obtained indicate that RF was suitable for landslide susceptibility mapping, there is no determined relationship between debris flow and landslide, it is feasible to map two kinds of disaster in the same susceptibility map (line 381-385).

5) English form is not always acceptable

Response: We have checked the whole manuscript again and avoid some awkward expressions.

6) Introduction The writer suggests a brief characterization of debris flows based on the triggering mechanisms and conditions to explain the phenomenon and avoid confusion,as that below: Most of debris flows are runoff generated (Imaizumietal. 2006; Coe et al.,2008, Gregoretti&DallaFontana, Ma et al.,2018). In many casesthey occur on a channel bed for the entrainment into abundant runoff of debris supplied by deep or shallow slides of slopes incised by the channel. (Theule et al.,(2012), Hurlimann et al.(2014), Imaizumi et al. 2019,Zhou et al.,.2019; Simoni et al.,2020).

Conversely,landslide or natural dam failure that evolve into a debris flow(Iverson et al.,; Kean et al.,2013) are not frequent. Moreover,it is not clear the way the potential relationship between debris flow and landslide is approached through the separated susceptibility analysis:some concise

information could help the reader.

Response: The comments are detailed and important and we have add related information in the Introduction to further express the potential relationships between debris flow and landslide (line 41,44-46, 48-49).

Other spotted errors and comments are as follows: Line75 perhaps "surface"instead of "area"

Response:We modified it accordingly. Line78

Line 82 "Three types of lithology were mainly observed"rather than"There were three common lithology observed"

Response:We modified it accordingly. Line85

Line85 "Main common disasters in the study area mainly consist"instead of"The disasters in the study area mainly consist"

Response:We have already modified it. Line88

Lines104-105"The geometry of debris flow is better represented by a polygon or a set of polygons in vector format"Which is the sense of such a sentence?Authors should explain,as in the case of landslide which typology of unit is preferable for debris flow.

Response:We have added related information. The watershed unit is preferable for debris flow. line 109-110.

Line115:Moreover, before availability

Response:We have already modified it (line 123).

Lines114-116"The occurrence of debris flow emphasizes the indispensability of provenience, topography and triggering factors. Availability, reliability, and practicality of thefactor data were also considered (van Westenetal.,2008)." Such period should be postponed to that at lines116-119.

Response:We have already modified it (line 122-124).

Line122:elevation of what?

Response: Maximum elevation difference is another conditioning factor.

line124:Figure 5 concerns landslide. The figure 6 concerning debris flow should be also added.The writer suggests to distinguish the reference to these two figures by means of the phenomenon.

Response:We have already modified it.

line126 please separate the controlling factors concerning landslide from those concerning debris flows.

Response:We have already modified it (Section 2.4.1-2.4.3).

Line127 Why do the authors use reclassify and not classify?

Response:Reclassify is an tool in ArcGIS platform.

Line128: the proximity of roads, rivers rather than roads, river....Therefore the distance from roads, river was classified

Response:We have already modified it (line 135-138).

Lines137-138"Considering the correlation between the two controlling factors,basin area and main channel length are represented by the same graph,which was reclassified into four classes(Fig.6h)."unclear sentence.

Response:We have already modified it. Line 148-149

Line148"Totally 18 factors are obtained by processing the row data in the ArcGIS10.2 platform."

Perhaps the values of 18 controlling factors were classified by processing......

Response: We have already modified it. Line 161-162

Line149 The DEM size,30 m seems too large.The author should justify it.Please con?sider that Boreggioetal.(2018) suggested the use of 1m grid size.

Response:The DEM size is accessible for 30m and 90m. Some studies use 5m or 1m by resampling tool of ArcGIS. However, 30m is the most common.

Line153"under study as a reference."Unclear expression.

Response:We have already modified it (line166-168).

Lines157-158"data set"rather than"set"

Response:We have already modified it.

Lines159-161The partition of landslide inventory is approached..........among them that of one time random selection() is the most used.

Response:We have already modified it (174-177).

Lines179-180"RF uses the bagging technique (bootstrap aggregation)to select, at each node of the tree,random samples of variables and observations as the training data set for model calibration."Unclear sentence

Response:We have already modified it (Line194-195).

Lines186-194.Unclear period.

Response:We have already modified it. Line 200-202.

Line197"training data set"rather than"training set"

Response:We have already modified it.

Line198"of for"????

Response: We have already modified it.

Line199"with a sensitivity value"

Response:We have already modified it.

Line203"for models"??????

Response:We have already modified it.

Line206 Values of 88.69 and 86.05% are claimed for sensitivity and specificity respec?tively.Why at the previous lines the values are 91.62 and 89.96%? Please explain

Response:The data set were divided into two groups, one for training, the other for validation. Therefore, the sensitivity and specificity values were different.

Line207"with a value"

Response:We have already modified it (line 214).

Line208"training model"???? perhaps it is training dataset

Response:We have already modified it (line 221).

Line214 delete"reached 179"

Response:We have already modified it .

Lines215,216,231 what does relate the percentage?The total number of units? Please Specify.

Response:We have already modified it (line 220, 230, 244).

Line219"were" in stead of "was"

Response:We have already modified it (line 233).

Line230 delete"reached to 26"

Response:We have already modified it.

Line235 which is the sense of the following sentence?'which has significant influence on the occurrence of debris flow.

Response:We have already modified it (line 249).

Line236 substitute "which are"with":"

Response:We have already modified it.

Figure8.The writer suggests the use of the same colours for both the susceptibility Maps.

Response: We have compared the results before and found it better when the    maps were made from different colour to highlight the difference between debris flow    and landslide.

Lines249-251"There are 23 watershed units belonging to high-class in the debris flow susceptibility zoning map (Fig.8),of which17 units are covered with high or very    high-class slope units in the landslide zoning map (Table5)." it is better substitute"are    covered" with "correspond to high or very high-class" Moreover, add the following sentence: "Therefore,there are 6 units that does not overlap (about26%)."

Response:We have already modified it (line 264-266).

Line251 about the 4 watershed units:do the belong to the 6 watershed units with no high or very-high slope units?

Response: The susceptibility maps were reclassified into five levels as very low, low, moderate, high and very high. And 17 watershed units were correspond to with high or very high-class slope units in the landslide zoning map, 4 watershed units are covered with low or very low class slope units. The last 2 watershed unis are covered with moderate level.

Lines252-253 which susceptibility maps do belong the 19 high and very-high class watershed units (19)

Response: Watershed units were for debris flow and slope units were for landslide.

Response: We have added detail information (line 274-275, 285).

Lines 313-314 "has been little used until now for susceptibility analysis of landslide and debris flows"instead of "has less application in landslide and debris flow analysis"

Response:We have already modified it (line 331-332).

Response:We have already modified it (343-347).

Lines326-329 this period should be summarized in a more concise and clear form

Response:We have already modified it (345-349).

Line 338 Where the relationship between landslide and debris flow is illustrated?

Response:We have added related information (line355-356).

Lines 348-349 "The fact that the appropriate prediction method and mapping units applied to the two disasters makes it possible to merge the two zoning maps"Which appropriate prediction method?Which is sense of this sentence?

Response: Random forest has proved its superiority in this study. Mapping two kinds of disaster in the same map has not been explored before and we try to explain why and how does it works.

Line 359 "models based on random forest"if the authors mean"based on RF models"the expression is unclear (seelines197;241).This ambiguity is elsewhere present in the submitted manuscript.

Response:We have already modified it (line 378).

Points 2 and 3 of the conclusion could be merged in a unique one.This point should begin after explaining that there is no potential relationship between the occurrence of the two considered phenomena. After that,the authors could explain the reasons in 2.1.and2.2 corresponding to the points 2 and 3 of the submitted work.

Response: We have modified it (line 381-385).

Finally, we have added related reference based on the comments.

Response: We have added related reference based on your list.

We appreciate for Editors and Reviews' warm work earnestly, and hope that the correction will

meet with approval.

Thank you and best regards.

Yours,

Zhu

---

## Referee Report (RR1)

**RE:** NHESS 2020 294R1        Liang et al. Exploring the potential relationship between the occurrence of debris flow and landslide

Unfortunately, this revised version is not acceptable for pubblication. The text is not clear and redundant. Moreover, in some parts the train of though seems missing. Grammar should be also revised. The reader stopped the revision at section 2.

The details below:

Debris flows are not landslides (see lines 11 and 13). Debris flows and landslides are gravitational mass transport phenomena. Moreover, statements at lines 13 ("An inventory map consisting of 448 landslides (399 soil slides and 49 debris flows") and 16 ("constructed for landslide and debris flow") are in contrast.

Lines 11-12 " occurred commonly" this is not an English form.

Lines 21 "with two kinds of disaster" this expression is not suitable, "with the two considered hazardous phenomena" could be better.

Line 22 "Two models" which models? "The two used models"?

Lines 23-25 "The loose sources need by the debris flow were not necessarily brought by the landslides although most landslides can be converted into debris flow. The area prone to debris flow did not promote the occurrence of landslide." Which is the sense or scope of this period? Moreover, are these outcomes from field surveys or from the model results? In the first case how are they related to susceptibility maps?

Lines 41-42 "Most of debris flows are runoff generated (Ma et al., 2018)." Such statement is misleading. Ma et al. (2018) do not state that most runoff are generated debris flows. In the previous review the writer suggested other references to confirm it. Therefore, at least the following references should be added: Imaizumi et al. (2006), Coe et al., (2008), Gregoretti and Dalla Fontana (2008), Theule et al. (2020).

Lines 44-46 "Debris flow usually occurs on a channel bed for the entrainment into abundant runoff of debris supplied by deep or shallow slides of slopes incised by the channel (Imaizumi et al.2019;

Zhou et al., 2019)" Again at least other two references should be added to provide a base to this statement: "Hurlimann et al. (2014) and Simoni et al. (2020)" .

Line 47 "and most of the slides are accompanied by debris flow" please add some reference

Lines 47-49 "In the past, it is not clear the way the potential relationship between debris flow and landslide is approached through the separated susceptibility analysis (Alessandro et al., 2015; Guzzetti et al., 2005)" Unclear period.

Lines 49-56 "In addition, some scholars made separate evaluations of slides and debris flow (Park et al., 2011; Haydar et al., 2016). Some scholars have proposed a coupled model of landslide-debris flow (Chiang et al., 2012; Gomes et al., 2013). However, not every slide has evolved into a debris flow and the material source of the debris flow is not necessary coming from slides. The formation and manifestations of different types of landslides are different, especially debris flow, which is a kind of "wet flow" (Varnes, 1978). In other words, there is no determined connection between debris flow and other types of landslide." Very confused and ill organized period. It should be rewritten in a more concise, clear and synthetic form. In addition, "not every slide has evolved into a debris flow" seems to contradict what written at line 47 "and most of the slides are accompanied by debris flow".

Lines 58-59 "Besides, the conditioning factors and mapping units involved in the susceptibility assessment different kinds of landslides are not identical." Another confused and unclear sentence.

Lines 61-62 "As an example, one landslide inventory map includes only one type of landslide, as does debris flow." Useless sentence: the same concept has been introduced at the previous line.

Lines 63-69 "The methods of susceptibility assessment can be broadly classified as qualitative or quantitative (Aleotti et al., 1999). Several methods and approaches have been proposed and tested to ascertain susceptibility, such as physical-based approaches (Carrara et al., 2008), heuristic methods (Blais et al., 2016) and statistically-based approaches (Reichenbach et al., 2018). In addition, new machine learning models, such as neural networks (Park et al.,2013), support vector machines (Colkesen et al.,2016) and random forest (RF) (Zhu et al., 2020a), have also been applied." This period is full of redundancy and as written does not merge with the text: it is not linked to the previous and following text. The following a proposal for rewriting it "The methods

used for the susceptibility assessment can be broadly classified as qualitative or quantitative (Aleotti et al., 1999). About the quantitative methods there are those physically-based (Carrara et al., 2008), those heuristic (Blais et al., 2016) and those statistically-based (Reichenbach et al., 2018). Recently new machine learning models have been used for susceptibility analysis: neural networks (Park et al.,2013), support vector machines (Colkesen et al.,2016) and random forest (RF) (Zhu et al., 2020a).

Lines 70-71 "The Longzi County in Southeastern Tibet is always exposed to slides and debris flow hazard because of climatic and topographic conditions, which is chosen as the study area The purpose of the present study is to explore the potential relationship between the occurrence of debris flow and soil slide by establishing susceptibility zoning maps separately with the use of random forest. It also provides a reference for the study of landslide-debris flow, a common disaster chain" Again all this period is not properly written and seems a collage of sentences, in the sense that a train of though is missing.

Line 83 "belongs" too many repetitions.

Lines 111-112 "First-order sub-catchments, which is also called watershed unit, was applied to the susceptibility of debris flow" sub-catchments is plural, therefore, it should be "are" and "were" instead of "is" and "was" respectively.

Lines 118-119 "there area lot of difference between the factors used by different landslide susceptibility assessments." Unclear sentence and "a" before "lot" is missing.

Lines 123-124 "Moreover, availability, reliability, and practicality of the factor data were also considered (van Westen et al., 2008)." Which is the sense of this sentence and its scope in the paper?

Line 119-130 All this period should be rewritten in a more organized and concise form. At the beginning it should be stated that 11 and 12 controlling factors are selected for landslide and debris flow susceptibility assessment respectively.

Line 142 "reclassified" why reclassified? Was it previously classified?

Line 148 "Basin area was reclassified into four classes and main channel length are represented" Unclear and grammatically incorrect sentence.

Line 158 "have" instead of "has".

Line 161 "The values of 18 controlling factors were classified by processing the raw data in the ArcGIS" At the previous lines 11 and 12 controlling factors are introduced for landslides debris flows respectively: please explain the new 18 controlling factors.

About rainfall: rainfall triggering debris flows is much different from those triggering landslides. The former is usually a short duration precipitation, while the latter is a long duration precipitations. Therefore, considering the annual rainfall depth for both the phenomena could not have a physical base.

Line 449 The reference is bad written: Francesco is a name, not a surname

REFERENCES

Coe, J. A., Kinner, D. A., and Godt, J. W. (2008). Initiation conditions for debris flows generated by runoff at Chalk Cliffs, Central Colorado. Geomorphology 96, 270–297. doi: 10.1016/j.geomorph.2007.03.017

Gregoretti, C., and Dalla Fontana, G. (2008). The triggering of debris fl ow due to channel - bed failure in some alpine headwater basins of the Dolomites: Analyses of critical runoff. Hydrological Processes, 22, 2248 – 2263.

Hurlimann M., Abanco C., Moya, J., Vilajosana I. 2013. Results and experiences gathered at the Rebaixader debris-flow monitoring site, Central Pyrenees, Spain. Landslides. doi:10.1007/s10346-013-0452-y 161-175

Imaizumi F, Sidle RC, Tsuchiya S, Ohsaka O. 2006. Hydrogeomorphic processes in a steep debris flow initiation zone. Geophysical Research Letters 33: L10404.

Simoni A., Bernard, M., Berti M., Boreggio M., Lanzoni S., Stancanelli L., Gregoretti C (2020) Runoff-generated debris flows: observation of initiation conditions and erosion-deposition dynamics along the channel at Cancia (eastern Italian Alps). Earth Surface Processes and Landforms - doi:10.1002/esp.4981

Theule, J.I., Liebault, F., Loye, A., Laigle, D., and Jaboyedoff, M., 2012. Sediment budget monitoring of debris flow and bedload transport in the Manival Torrent, SE France.

---

## Author Response (AR2)

Dear Editors and Reviewers

Thank you for your letter and for the Reviewers' comments concerning our manuscript entitled "Exploring the potential relationship between the occurrence of debris flow and landslide" (ID: NHESS-294). Those comments are all valuable and very helpful for revising and improving our paper, as well as the important guiding significance to our researches. We have studied comments carefully and have made correction which we hope meet with approval. There is another thing we want to discuss with you that the Acknowledgement information need to be revised as "This work was supported by Graduate Innovation Fund of Jilin University and the National Natural Science Foundation of China (Grant No. 41972267, 41977221, and 41572257) ." Revised portion are marked in red in the paper. The main corrections in the paper and the responds to the Reviewer's comments are as flowing:

For the first reviewer:

This paper presents an interesting comparison between landslide and debris flow susceptibility maps generated using Random Forest Classification. The majority of the corrections I suggested in my previous review have been carried out, and I only have the following brief comments to add, which I think will improve the clarity of the manuscript:

Response: Thank you for your approval of our work.

1. Line 17 - ROC curve stands for receiver operating characteristic, not relative operating characteristic

Response: We are agreed with you and have made it corrected.On Line 17

2. Line 23 - The order of this sentence makes it slightly unclear, it might be clearer to give to ROC

for the two models and then the accuracy

Response: We are agreed with you and have made it corrected. On Line 21

3. Line 126 - I think you should specify here that you are referring to pre-event NDVI, since the

change in NDVI is also often used to detect landslides

Response: We are agreed with you and have made it clear. On Line 122

4. Line 212 - Should specificity also be percentage?

Response: Yes, it should be percentage. On Line 189

5. Line 264 - remove "with"

Response: We are agreed with you and have made it clear.

6. Line 250 - Is "Melton" used throughout the paper to refer to the melton ratio, as defined on line

145? It is also used on line 126 before it has been described. I think that where it is used in tables

and figures, it would be better to refer to "melton ratio" than "melton" - it will not take up more

space since the name is short and might help people to understand.

Response: We are agreed with you and have made it clear. On line 126, 227.

7. Line 277 - Define acronym SPSS

Response: We are agreed with you and have made it clear. On line 254.

8. Line 381 - you say there is no relationship between the occurrence of landslides and debris flow. I think this should be reworded as in the next sentence, you explain that landlslides can be converted into debris flow, which would imply that there is a relationship. Perhaps you mean that the relationship is not straightfoward? Or that the models of landslide and debris flow are not interchangeable?

Response: Yes, the relationship is not that straightforward. We have reworded this conclusion. On line 358.

For the second reviewer:

Unfortunately, this revised version is not acceptable for pubblication. The text is not clear and redundant. Moreover, in some parts the train of though seems missing. Grammar should be also revised. The reader stopped the revision at section 2.

Response: We are sorry for all the mistakes we make and we will revise the manuscript based on your comments.

1. Debris flows are not landslides (see lines 11 and 13). Debris flows and landslides are gravitational mass transport phenomena. Moreover, statements at lines 13 ("An inventory map consisting of 448 landslides (399 soil slides and 49 debris flows") and 16 ("constructed for landslide and debris flow") are in contrast.

Response: Yes, these please are unclear and we have revised the statements.On line12,18.

2. Lines 11-12 " occurred commonly" this is not an English form.

Response: It should be occurred frequently.

3. Lines 21 "with two kinds of disaster" this expression is not suitable, "with the two considered hazardous phenomena" could be better.

Response: We are agreed with you and have made it corrected. On line 21

4. Line 22 "Two models" which models? "The two used models"?

Response: Two models represent for landslide and debris flow susceptibility models constructed by random forest. It should be two used models. On line 22

5. Lines 23-25 "The loose sources need by the debris flow were not necessarily brought by the landslides although most landslides can be converted into debris flow. The area prone to debris flow did not promote the occurrence of landslide." Which is the sense or scope of this period? Moreover, are these outcomes from field surveys or from the model results? In the first case how are they related to susceptibility maps?

Response: All the conclusion we obtained in the study come from the model results. We should first present the combined maps and then the conclusion. It will be more clear and we have added related information. On line 23-25.

6. Lines 41-42 "Most of debris flows are runoff generated (Ma et al., 2018)." Such statement is misleading. Ma et al. (2018) do not state that most runoff are generated debris flows. In the

previous review the writer suggested other references to confirm it. Therefore, at least the following references should be added: Imaizumi et al. (2006), Coe et al., (2008), Gregoretti and Dalla Fontana (2008), Theule et al. (2020).

Response: We have added the related references. On line 41-42

7.  Lines 44-46 "Debris flow usually occurs on a channel bed for the entrainment into abundant runoff of debris supplied by deep or shallow slides of slopes incised by the channel (Imaizumi et al.2019; Zhou et al., 2019)" Again at least other two references should be added to provide a base to this statement: "Hurlimann et al. (2014) and Simoni et al. (2020)" .

Response: We have added the related references.On line 46-47.

8.  Line 47 "and most of the slides are accompanied by debris flow" please add some reference

Response: We have added the related references.On line 48-49.

9.  Lines 47-49 "In the past, it is not clear the way the potential relationship between debris flow and landslide is approached through the separated susceptibility analysis (Alessandro et al., 2015; Guzzetti et al., 2005)" Unclear period.

Response: It should be "In the past, seldom researches have explored the potential relationship between debris flow and landslide through the separated susceptibility maps (Alessandro et al., 2015; Guzzetti et al., 2005) " line 49-56

10. Lines 49-56 "In addition, some scholars made separate evaluations of slides and debris flow

(Park et al., 2011; Haydar et al., 2016). Some scholars have proposed a coupled model of landslide-debris flow (Chiang et al., 2012; Gomes et al., 2013). However, not every slide has evolved into a debris flow and the material source of the debris flow is not necessary coming from slides. The formation and manifestations of different types of landslides are different, especially debris flow, which is a kind of "wet flow" (Varnes, 1978). In other words, there is no determined connection between debris flow and other types of landslide." Very confused and ill organized period. It should be rewritten in a more concise, clear and synthetic form. In addition, "not every slide has evolved into a debris flow" seems to contradict what written at line 47 "and most of the slides are accompanied by debris flow".

Response: We agree with the comment. Debris flow and landslide should be expressed individually. We have rewritten the part. "not every slide has evolved into a debris flow" seems to contradict what written at line 47 "and most of the slides are accompanied by debris flow". The statement is redundant.    line 49-57

11. Lines 58-59 "Besides, the conditioning factors and mapping units involved in the susceptibility assessment for different kinds of landslides are not identical." Another confused and unclear sentence.

Response: It should be "Besides, the conditioning factors and mapping units involved in the susceptibility assessment for different kinds of landslides are not identical."

12. Lines 61-62 "As an example, one landslide inventory map includes only one type of landslide, as does debris flow." Useless sentence: the same concept has been introduced at the previous

line.

Response: We have deleted the sentence.

13. Lines 63-69 "The methods of susceptibility assessment can be broadly classified as qualitative or quantitative (Aleotti et al., 1999). Several methods and approaches have been proposed and tested to ascertain susceptibility, such as physical-based approaches (Carrara et al., 2008), heuristic methods (Blais et al., 2016) and statistically-based approaches (Reichenbach et al., 2018). In addition, new machine learning models, such as neural networks (Park et al.,2013), support vector machines (Colkesen et al.,2016) and random forest (RF) (Zhu et al., 2020a), have also been applied." This period is full of redundancy and as written does not merge with the text: it is not linked to the previous and following text. The following a proposal for rewriting it "The methods used for the susceptibility assessment can be broadly classified as qualitative or quantitative (Aleotti et al., 1999). About the quantitative methods there are those physically-based (Carrara et al., 2008), those heuristic (Blais et al., 2016) and those statistically-based (Reichenbach et al., 2018). Recently new machine learning models have been used for susceptibility analysis: neural networks (Park et al.,2013), support vector machines (Colkesen et al.,2016) and random forest (RF) (Zhu et al., 2020a)

Response: Thank you for the proposal and the sentences seem more logical and clear. Line 58-63

14. Lines 70-71 "The Longzi County in Southeastern Tibet is always exposed to slides and debris flow hazard because of climatic and topographic conditions, which is chosen as the study area The purpose of the present study is to explore the potential relationship between the

occurrence of debris flow and soil slide by establishing susceptibility zoning maps separately with the use of random forest. It also provides a reference for the study of landslide-debris flow, a common disaster chain" Again all this period is not properly written and seems a collage of sentences, in the sense that a train of though is missing.

Response: We have rewrote this paragraph. Line 64-67

15. Line 83 "belongs" too many repetitions.

Response: We used be distributed in. Line 75

16. Lines 111-112 "First-order sub-catchments, which is also called watershed unit, was applied to the susceptibility of debris flow" sub-catchments is plural, therefore, it should be "are" and "were" instead of "is" and "was" respectively.

Response: Yes, thank you so much. We have revised them.   line 100-103.

17. Lines 118-119 "there area lot of difference between the factors used by different landslide susceptibility assessments." Unclear sentence and "a" before "lot" is missing.

Response: We are sorry for the mistake. It should be "there are a lot of difference between the factors used by different landslide susceptibility assessments." But we have revised this sentence. Line 106-110.

18. Lines 123-124 "Moreover, availability, reliability, and practicality of the factor data were also considered (van Westen et al., 2008)." Which is the sense of this sentence and its scope in the

paper?

Response: It should be presented firstly. We have revised this sentence on Line 106-110.

19. Line 119-130 All this period should be rewritten in a more organized and concise form. At the beginning it should be stated that 11 and 12 controlling factors are selected for landslide and debris flow susceptibility assessment respectively.

Response: Yes, this paragraph is obviously lengthy. We have deleted the duplicate information. Line 106-110.

20. Line 142 "reclassified" why reclassified? Was it previously classified?

Response: Reclassify is an tool in ArcGIS platform. Controlling factors belong to continuous variable (like aspect, rainfall..). Reclassify helps to express the difference between different units.

21. Line 148 "Basin area was reclassified into four classes and main channel length are represented" Unclear and grammatically incorrect sentence.

Response: We have revised this sentence. Line 128-129.

22. Line 158 "have" instead of "has".

Response: We have revised this sentence. Line 138-139.

23. Line 161 "The values of 18 controlling factors were classified by processing the raw data in the ArcGIS" At the previous lines 11 and 12 controlling factors are introduced for landslides

debris flows respectively: please explain the new 18 controlling factors.

Response: Not the new 18 controlling factors. There are 11 and 12 controlling factors are introduced for landslides debris flows respectively. But 4 factors like rainfall, slope, Maximum elevation difference and elevation were both used in landslide and debris flow. "18" have been deleted in the text.

24. About rainfall: rainfall triggering debris flows is much different from those triggering landslides. The former is usually a short duration precipitation, while the latter is a long duration precipitations. Therefore, considering the annual rainfall depth for both the phenomena could not have a physical base.

Response: Rainfall as the only triggering factor in the paper should be considered in landslide and debris flow. We agree with the comment that the mechanism of induction may different between debris flow and landslide. Rainfall in the study are mainly concentrated on summer. Annual rainfall has also been applied for many times. It is true that both intensity and persistence of rainfall have the influence on the occurrence of landslide and debris flow.

25. Line 449 The reference is bad written: Francesco is a name, not a surname

Response: We have revised it.

We appreciate for Editors and Reviews' warm work earnestly, and hope that the correction will meet with approval. Thank you and best regards.

Yours,

Zhu